**Multi-source hydrological soil moisture state estimation using data fusion optimisation**
Lu Zhuo[1*], Dawei Han[1]
[1]WEMRC, Department of Civil Engineering, University of Bristol, Bristol, UK
[*]Correspondence: lu.zhuo@bristol.ac.uk
**Abstract**
Reliable estimation of hydrological soil moisture state is of critical importance in operational
hydrology to improve the flood prediction and hydrological cycle description. Although there
have been a number of soil moisture products, they cannot be directly used in hydrological
modelling. This paper attempts for the first time to build a soil moisture product directly
applicable to hydrology using multiple data sources retrieved from SAC-SMA (soil moisture),
MODIS (land surface temperature), and SMOS (multi-angle brightness temperatures in H-V
polarizations). The simple yet effective Local Linear Regression model is applied for the data
fusion purpose in the Pontiac catchment. Four schemes according to temporal availabilities of
the data sources are developed, which are pre-assessed and best selected by using the well-
proven feature selection algorithm Gamma Test. The hydrological accuracy of the produced
soil moisture data is evaluated against the Xinanjiang hydrological model's soil moisture
deficit simulation. The result shows that a superior performance is obtained from the scheme
with the data inputs from all sources ($NSE = 0.912$, $r = 0.960$, $RMSE = 0.007$ m). Additionally
the final daily-available hydrological soil moisture product significantly increases the Nash-
Sutcliffe efficiency by almost 50 % in comparison with the two most popular soil moisture
products. The proposed method could be easily applied to other catchments and fields with
high confidence. The misconception between the hydrological soil moisture state variable and
the real-world soil moisture content, and the potential to build a global routine hydrological
soil moisture product are discussed.
**Keywords**: Hydrological soil moisture state (SMD); Local Linear Regression (LLR); Gamma
Test (GT); Soil Moisture and Ocean Salinity (SMOS) multi-angle brightness temperatures;
North American Land Data Assimilation System 2 (NLDAS-2); Moderate Resolution Imaging
Spectroradiometre (MODIS) land surface temperature

## 1. Introduction

Soil moisture is a key element in the hydrological cycle, regulating evapotranspiration,
precipitation infiltration and overland flow (Wanders et al., 2014). For hydrological
applications, the antecedent wetness condition of a catchment is among the most significant
factors for accurate flow generation processes (Berthet et al., 2009; Matgen et al., 2012a).
(Norbiato et al., 2008) reported that initial wetness conditions are essential for efficient flash
flood alerts. Additionally an operational system requires reliable hydrological soil moisture
state updates to reduce the time drift problem (Aubert et al., 2003; Berg and Mulroy, 2006;
Dumedah and Coulibaly, 2013). However, currently there is no available soil moisture product
that can be used directly in hydrology modelling, primarily because soil moisture is difficult to
define and there is no single shared meaning in various disciplines (Romano, 2014).
Although there have been many soil moisture measuring projects (e.g., satellite missions such
as Advanced Scatterometer (ASCAT), Soil Moisture and Ocean Salinity (SMOS), and Soil
Moisture Active Passive (SMAP); ground-based networks such as Soil Climate Analysis
Network (SCAN), U.S. Surface Climate Observing Reference Networks (USCRN), and
COsmic-ray Soil Moisture Observing System (COSMOS)), they are not sufficiently used in
hydrology due to the following reasons: 1) misconception between the hydrological soil
moisture state variable and the real-field soil moisture content (Zhuo and Han, 2016a); 2)
unawareness of data availability and strength/weakness of different data sources; 3) the existing
soil moisture products are mainly evaluated against point-based ground soil moisture
observations or airborne retrievals which have significant spatial mismatch (both horizontally
and vertically) to catchment-scales, and are therefore less applicable to hydrological modelling
(Pierdicca et al., 2013); 4) underutilisation of multiple data sources (e.g., multi-angle raw
observations by satellite sensors).
Some studies have attempted to directly utilise the existing soil moisture products (i.e., data
from satellites, land surface models, and in-situ methods directly) for flood prediction
improvement, for example (Brocca et al., 2010) explored that utilising the soil water index
from ASCAT sensor could improve runoff prediction mainly if the initial catchment wetness
conditions were unknown; (Aubert et al., 2003) assimilated in-situ soil moisture observations
into a simple rainfall-runoff model and acquired better flow prediction performance ; (Javelle
et al., 2010) suggested that estimations of antecedent soil moisture conditions were useful in
improving flash flood forecasts at ungauged catchments; contrarily (Chen et al., 2011)'s study
showed assimilating ground-based soil moisture observations was generally unsuccessful in
enhancing flow prediction; and (Matgen et al., 2012b) revealed that satellite soil moisture
products added little or no extra value for hydrological modelling. Clearly those results are
rather mixed. Challenges remain in integrating soil moisture estimated outside the hydrological
field into hydrological models. We believe if a hydrologically directly applicable soil moisture
product could be produced, the aforementioned studies' results would be significantly
improved.
Therefore the aims of this paper are to clarify the aforementioned misconception between the
hydrological model's soil moisture state and the real-world soil moisture, assess the data
availabilities for direct hydrological soil moisture state estimation, and fuse those available
data sources using a hydrologically relevant approach. It is hoped that the final product has a
superior hydrological compatibility over the existing soil moisture products. To achieve these
aims, the Xinanjiang (XAJ) (Zhao, 1992) operational rainfall-runoff model is used as a target
to simulate flow and soil moisture state information (i.e., soil moisture deficit (SMD)) for the
Pontiac catchment in the central United States (U.S.). The reason for adopting XAJ is explained
in the following section. For the purpose of hydrological soil moisture state estimation, it is
effective to adopt the data driven method, which can map multiple data sources into the desired
dataset without computational burden. In this study the Local Linear Regression (LLR) model
is used. The multiple data sources applied in this study include the SMOS (Kerr et al., 2010b)
multi-angle brightness temperatures ($T_b$s) with both horizontal (H) and vertical (V)
polarizations, the Moderate Resolution Imaging Spectroradiometre (MODIS) (Wan, 2008)
land surface temperature, and the soil moisture product by SAC-SMA (Xia et al., 2014). The
detail explanations of those datasets are covered in the methodology section. A well-proven
feature selection algorithm Gamma Test (GT) (Stefánsson et al., 1997; Zhuo et al., 2016b) is
employed to pre-assess the selected data inputs and find the optimal combination of them for
soil moisture state calculation. In addition, an *M*-test (Remesan et al., 2008) is adopted to
explore the best size of the training data. The desired soil moisture product is trained and tested
by the XAJ SMD simulation. In total four data-input schemes are developed according to the
temporal availability of the selected data inputs, which are then combined to give a daily
hydrological soil moisture product.
Compared with previous work, our study contains the following new elements: i) a
hydrologically directly usable soil moisture product is proposed; ii) the GT and LLR techniques
are used for the first time in a data fusion of multiple data sources for hydrological soil moisture
state estimation; iii) the use of multiple data sources is useful, which allows data users to
analyse the availability of the different products and compare the relative benefits of them.
**2.  Material and Methods**
**2.1 Study Area**
In this study, the Pontiac catchment (1,500 km$^2$, Figure 1) is used for the calibration and the
validation of the XAJ model. Pontiac (40.878°N, 88.636°W) lies on the north-flowing
Vermilion River, which is a tributary of the Illinois River of the state of Illinois, U.S. The worst
flood in this area occurred on December 4, 1982, cresting at 5.84 m above mean sea level
(MSL); and the most recent flood occurred on January 9, 2008, cresting at 5.75 m MSL, so this
catchment is likely located within a winter-flooding region. Pontiac is covered with moderate
canopy (the annual mean Normalized Difference Vegetation Index retrieved from the MODIS
satellite is around 0.4), when compared with a densely vegetated catchment, it has more
accurate soil moisture estimations from satellites (Al-Bitar et al., 2012). Based on the Köppen-
Geiger climate classification, this medium sized catchment is dominated mainly by hot summer
continental climate (Peel et al., 2007). With reference to the University of Maryland
Department Global Land Cover Classification, it is used primarily for agriculture purpose
(Bartholomé and Belward, 2005; Hansen, 1998). The soil mostly consists of Mollisols, which
has deep and high organic matter, and the nutrient-enriched surface soil is typically between
60-80 cm in depth (Webb et al., 2000). The study period is from January 2010 to December
2011. The reason for using this two-year period of data is because there have been many data
gaps from 2013-2017, and the data quality in 2012 was poor. As a result, only the data in 2010-
2011 are consistent and of high quality. As pointed out by Liu and Han 2010, 'Traditionally,
hydrologists use rules of thumb to select a certain period of hydrological data to calibrate the
models (i.e., 6 year data).' However, their study has shown 'the information content of the
calibration data is more important than the data length; thus 6 month data may provide more
useful information than longer data series.' Therefore, the two years of high quality data
adopted in the study are better than a longer period of poor quality data.
The North American Land Data Assimilation System 2 (NLDAS-2) (Mitchell et al., 2004)
provides precipitation and potential evapotranspiration information to run the XAJ model. Both
data forces are at $0.125°$ spatial resolution and have been converted to daily temporal resolution.
In order to use those distributed forcing into the lumped XAJ model, both forcing have been
interpolated with the area-weighted average method instead of the more complicated Kriging
approach, because the latter could produce errors if not well controlled (Wanders et al., 2014).
The average annual rainfall depth is about 954 mm, and the average annual potential
evapotranspiration is approximately 1670 mm. The daily observed flow data are acquired from
the U.S. Geological Survey.
**2.2 Hydrological Model**
The XAJ hydrological model is used for the simulation of SMD and river flow at a daily time
step. It is a simple lumped rainfall-runoff model with many applications performed in world-
wide catchments (Chen et al., 2013; Gan et al., 1997; Shi et al., 2011; Zhao, 1992; Zhao and
Liu, 1995; Zhuo et al., 2016a; Zhuo et al., 2015b). Since XAJ can obtain rather effective flow
modelling performances and require only two meteorological forcing (precipitation and
potential evapotranspiration) inputs (Peng et al., 2002), it is used more widely than the more
complicated  semi-distributed/  fully-distributed  hydrological  models  for  operational
applications.
As shown in Figure 2, the XAJ model has three main components: evapotranspiration, runoff
generation, and runoff routing. XAJ consists of soil layers (upper, lower and deep) in its
evapotranspiration calculations. Because XAJ adopts the multi-bucket variable-size method in
its modelling concept, it has unfixed soil depths which is more effective than the fixed depths
models (Beven, 2012). Other widely used models such as PDM (Moore, 2007), VIC (Liang et
al., 1994), and ARNO (Todini, 1996) also follow this concept.
In XAJ, the three-layer soil moisture state variables are all calculated as SMD, which is an
important soil wetness variable in hydrology. SMD is defined as the amount of water to be
added to a soil profile to bring it to the field capacity (Calder et al., 1983; Rushton et al., 2006).
In this study, only the surface SMD (i.e., top layer) referring to the vegetation and the very thin
topsoil, is utilised as a hydrological soil moisture target. This is because the water held in the
top few centimetres of the soil has been widely recognised as a key variable associated with
water fluxes (Eltahir, 1998; Entekhabi and Rodriguez-Iturbe, 1994). Moreover the current
satellite technology is only capable of acquiring the Earth information from the outermost layer
of the soil. Therefore as a case study based on the XAJ model, we only focus on the surface
soil moisture state investigation here. Future research will focus on the root-zone soil moisture
product development by using a similar method proposed in this study.
In this study, a modified version of the XAJ model is adopted, and interested readers are
referred to (Zhuo and Han, 2016b) for more details. All the XAJ's 17 parameters are used
during the model calibration, which are shown in Table 1. In this study, the genetic algorithm
(Wang, 1991) is used for parameter optimisation. Based on the genetic algorithm result, minor
trial and error adjustments to the parameters *EX*, *B*, *WUM*, *WLM* and *WDM* are also carried out
to obtain the best model performance (Chen and Adams, 2006). The calibration and the
validation results (during January 2010-April 2011 and May 2011 to December 2011,
respectively) of the XAJ model can be found in (Zhuo et al., 2015a). Discussion regarding the
river flow and SMD simulation results in this catchment have been published in (Zhuo and
Han, 2016b), with Nash-Sutcliffe Efficiency (*NSE*) obtained larger than 0.80 during both the
calibration and validation periods. The results are not repeated here.
**2.3 Multiple Data Sources for Hydrological Soil Moisture State Estimation**
Data sources from SMOS, MODIS and SAC-SMA are used (Table 2). All data sources have
been converted into catchment-scale datasets by the area-weighted average method. The detail
description of each data source is given as follows. The main reason for choosing those three
data sources is due to their Near-Real-Time (NRT) availabilities (MODAPS Services, 2015;
Rodell, 2016) (SMOS becomes available in NRT recently (ESA Earth Online, 2016)), which
allows fast implementation in flood forecasting.

**2.3.1 SMOS Multi-angle Brightness Temperatures (SMOS-T$_b$s)**

The SMOS (1.4 GHz, L-band) Level-3 T$_b$s data covering the studying period are available from
the Centre Aval de Traitement des Données SMOS (CATDS) (Jacquette et al., 2010). The
reason for choosing the SMOS satellite is because compare with other satellite techniques (i.e.,
optical, and thermal infrared), microwave bands (especially with longer wavelength such as L-
band (21 cm)) can penetrate deeper into the soil (~ 5 cm) and have less interruptions from
weather conditions (Njoku and Kong, 1977). Additionally SMOS has a relatively longer period
of data record compares with other satellite missions such as SMAP. SMOS retrieves the
thermal emission from the Earth in both H and V polarizations with a wide ranges of incidence
angles from $0^o$ to $60^o$. The observation depth of SMOS is approximately 5 cm with a spatial
resolution of 35-50 km depending on the incident angle and the deviation from the satellite
ground track (Kerr et al., 2012; Kerr et al., 2010a; 2001).
SMOS provides T$_b$s retrievals at all incidence angles averaged in $5^o$ -width angle bins, which
have been transformed into the ground polarization reference frame (i.e., H, and V
polarizations). Therefore the number of the SMOS-T$_b$s inputs for the hydrological soil moisture
estimation can be as high as 24 (12 angle bins per polarization), with the centre of the first
angle bin at $2.5^o$ in both polarizations (Rodriguez-Fernandez et al., 2014). As satellite
progresses, any given location on the Earth's surface is scanned a number of times at various
incidence angles, depending on the location with respect to the satellite subtrack: the further
away, the fewer the angular acquisitions (Kerr et al., 2010b). The data availabilities of the
SMOS-$T_b$s are illustrated in Figure 3 (the availabilities for H and V polarizations are the same).
It can be seen that the data availabilities among various incidence angles are rather different.
In this study the only angle range that gives the most available record of data is from 27.5° to
57.5° (i.e., 7 for H and 7 for V polarization), which is therefore chosen for the hydrological soil
moisture development. This angle range is in line with the angle selection in (Rodriguez-
Fernandez et al., 2014). In addition the SMOS Level-3 soil moisture product from the CATDS
(SMOS-SM) is also acquired for a comparison with the estimated soil moisture product.
Retrievals that are potentially contaminated with Radio Frequency Interference have been
removed. Readers are referred to (Kerr et al., 2012) for a full description of the SMOS
retrieving algorithms, and (Njoku and Entekhabi, 1996) for a good knowledge of how passive
microwave relates to soil moisture variations.
**2.3.2 MODIS Land Surface Temperature (MODIS-LST)**
The MODIS/Terra (Earth Observing System AM-1 platform) (Wan, 2008) daily MOD11C1-
V5 land surface temperature covering the studying period is downloaded from the Land
Processes Distributed Active Archive Centre website. MODIS is chosen among other
operational optical satellites for its suitable features, mostly, due to its frequent revisiting time
and free NRT data availability. It measures 36 spectral bands between 0.405 and 14.385 μm,
and acquires data at three spatial resolutions 250 m, 500 m, and 1,000 m respectively while the
adopted MOD11C1 V5 product incorporates 0.05º (5.6 km) spatial resolution. The benefit of
adding land surface temperature information is that previous studies have shown the variations
in soil moisture have a strong linkage with land surface temperature (Carlson, 2007; Goward
et al., 2002; Mallick et al., 2009). One reason is the changes of land surface temperature are
mainly affected by albedo and diurnal heat capacity, and the diurnal heat capacity is mainly
controlled by soil moisture (Price, 1980). (Wan, 2008) compared MOD11C1-V5 land surface
temperatures in 47 clear-sky cases with in situ measurement and revealed that the accuracy was
better than 1 K in the range from −10° to 58 °C in about 39 cases. Cloud-contaminated data
have been removed by a double-screening method, and its detail can be found in (Wan et al.,

222    2002).

**2.3.3 SAC-SMA Soil Moisture Estimation (SAC-SMA-SM)**
The reason for choosing the SAC-SMA land surface modelled soil moisture product is because
satellite can often have missing data due to various weather and canopy conditions (e.g., rainfall,
frozen weather, and vegetation coverage), so this daily dataset is essential in producing a
temporally completed hydrological soil moisture product. In this study, the surface soil
moisture (0-10 cm) simulated from the SAC-SMA model is selected. This is because its
estimated soil moisture gives a high accuracy against the observational soil moisture and a
good correlation with the XAJ SMD (Zhuo et al., 2015b). The daily SAC-SMA-SM is given
in   a   spatial   resolution   of   0.125º.   The   dataset   can   be   download   from
(http://www.emc.ncep.noaa.gov/mmb/nldas/ ). Readers are referred to (Xia et al., 2012) for a
full description of the SAC-SMA data products.

### 2.3.4 Data Availabilities

As shown in Table 2, the availability of the three data sources is rather different. Unlike SMOS

and MODIS, SAC-SMA-2 SM is a model based product which runs in a NRT mode, so it

produces valid data every day during the whole studying period. Whereas the two satellites'

data are more exiguous depends on weather and surface conditions. Compared with MODIS,

the SMOS's retrieval is even sparse and the biggest data shortage normally occurs in the winter

season where its returned microwave signal is mostly affected by frozen soils (Zhuo et al.,

2015a). Based on the data availability analysis, the proposed hydrological soil moisture product

is built from four data-input schemes as presented in Table 3. Those four schemes enable us to

test and compare the estimated soil moisture state more comprehensively. Since the continuity

of a soil moisture product is essential for any operational applications, SAC-SMA-SM is

included in all of the schemes.

### 2.4 Data Fusion

### 2.4.1  Gamma Test (GT) for Feature Selection

Before model building, it is important to carry out a feature selection process, because it can

simplify the model inputs, shorter training times, and reduce overfitting problems. In this study

a proper combination of the incidence angles from the SMOS $T_b$s is vital for the best soil

moisture state calculation. For this purpose, a feature selection method called GT is adopted. It

has been effectively used in numerous studies for model inputs selection (Durrant, 2001; Jaafar

and Han, 2011; Noori et al., 2011; Remesan et al., 2008; Tsui et al., 2002; Zhuo et al., 2016b).

In addition to the feature selection, GT can also give useful indication about the underlying

model complexity. It is a near-neighbour data analysis routine which determines the minimum
mean-squared error (*MSE*) that can be achieved based on the input-output dataset utilising any
continuous nonlinear models (Zhuo et al., 2016b). The calculated minimum *MSE* is referred as
the Gamma statistics and denoted as $\Gamma$. For detailed calculations about the GT algorithm,
interested readers are referred to (Koncar, 1997; Pi and Peterson, 1994; Stefánsson et al., 1997).
Here only the basic knowledge about the GT is shown:
$\{(x_i, y_i),\ 1 \le i \le M\}$                                    (1)
here the inputs $x_i \in R^m$ are vectors restricted by a closed bounded set $C \in R^m$, and their
corresponding outputs $y_i \in R$ are scalars, *M* stands for the sample points. The outputs *y* are
determined by the input vectors *x* that carry predictively useful messages. The only assumption
made is that their latent relationship is from the following function:
$y = f(x_1 \ldots x_m) + r$                                    (2)
here *f* is built up as a smooth model with *r* representing random noise. Without loss of generality,
the assumption of *r* noise distribution is that its mean is always zero, because all the constant
bias has been considered within the *f* model. Additionally *r*'s variance ($Var(r)$) is restricted
within a set boundary. The observations' potential model is now defined within the class of
smooth functions.
The $\Gamma$ is related to $N[i,k]$, which represents as the *k*th ($1 \le k \le p$) nearest neighbours of each
vector $x_i$ ($1 \le i \le M$), written as $x_{N[i,k]}(1 \le k \le p)$, where *p* is a fixed integer. In order to
determine the Gamma function from the input vectors, the *Delta* function is used:
$$\delta_M(k) = \frac{1}{M}\sum_{i=1}^{M}\left|x_{N[i,k]} - x_i\right|^2 \qquad (1 \leq k \leq p) \qquad\qquad (3)$$
here the function $\left|x_{N[i,k]} - x_i\right|$ calculates the Euclidean distance. The Gamma function for its
output values is expressed as in Eq. 4, and the $\Gamma$ can be determined from Eq. 3 and 4:
$$\gamma_M(k) = \frac{1}{2M}\sum_{i=1}^{M}\left|y_{N[i,k]} - y_i\right|^2 \qquad (1 \leq k \leq p) \qquad\qquad (4)$$
here $y_{N[i,k]}$ is the corresponding output values for the $k$th nearest neighbours $x_i$ ($x_{N[i,k]}$). To
find $\Gamma$ a least-squared regression line for the $p$ points ($\delta_M(k)$, $\gamma_M(k)$) is built using the
following equation:
$$\gamma = A\delta + \Gamma \qquad\qquad (5)$$
where $\Gamma$ can be determined when $\delta$ is set as zero. The detailed explanation is:
$$\gamma_M(k) \rightarrow Var(r), \text{ when } \delta_M(k) \rightarrow 0 \qquad\qquad (6)$$
Eq. 5 gives us valuable information about the underlying system: not only that the $\Gamma$ is a useful
indicator of the optimal *MSE* result that any smooth functions can achieve, but its gradient *A*
also provides guidance about the underlying model complexity (i.e., the steeper the gradient
the more sophisticated the model should be adopted). In this study, the winGamma$^{TM}$ software
is used for GT calculation (Durrant, 2001). The mathematical feasibility of GT has been
published in (Evans and Jones, 2002).

### 2.4.2 *M*-test for Training Data Size Selection

A common practice in nonlinear modelling is to split the dataset into training and testing parts.
However there is no universal solution on how to divide the datasets (i.e., the proportion of
each part) so that the best modelling results could be obtained. Here, an $M$-test is carried out,
where $M$ stands for the training data size. $M$-test is accomplished by calculating the $\Gamma$ for
increasing the $M$ value (i.e., expanding the training data) and exploring the resultant graph to
judge whether the $\Gamma$ approaches a stable asymptote. Such an approach is straightforward and
effective in finding the optimal sizes of training and testing datasets, while avoiding overfitting
problems and reducing unsystematic attempts.

### 2.4.3  Local Linear Regression (LLR)

Various data fusion techniques have been developed (Prakash et al., 2012; Srivastava et al.,
2013; Wagner et al., 2012), however their methods require high computational time to run and
this, in a real-time flood forecasting framework, could not match the operational needs.
Comparatively, LLR model is a simpler method and requires relatively low computational time.
Therefore it is chosen in order to test if a simple method is able to provide effective
performance. LLR is a nonparametric regression model that has been applied in (Liu et al.,
2011; Pinson et al., 2008; Sun et al., 2003; Zhuo et al., 2016b) for forecasting and smoothing
purposes. LLR builds local linear regression based on the nearest points ($p_{max}$) of a targeted
point, and repeats such a process over the whole training dataset to produce a piecewise linear
model. There are many methodologies in selecting the $p_{max}$, in this study a method called
influence statistics is used (Durrant, 2001; Remesan et al., 2008), which is outlined as below.
Assume there are $p_{max}$ nearest points, then the Eq. 7 can be built:
$Xm = y$               (7)
here $X$ is a $p_{max} \times d$ matrix which shows the $d$ dimensional information of $p_{max}$, $x_i$ are the
nearest points confined between 1 and $p_{max}$, $y$ is the output vector with $p_{max}$ dimension, and $m$
is a set of parameters formed in a vector, which plays an important role in mapping the solution
from $X$ to $y$. Therefore Eq. 7 can be expanded as
$$
\begin{pmatrix}
x_{11} & x_{12} & x_{13} & \cdots & x_{1d} \\
x_{21} & x_{22} & x_{23} & \cdots & x_{2d} \\
\vdots & \vdots & \vdots & \ddots & \vdots \\
x_{p_{max}1} & x_{p_{max}2} & x_{p_{max}3} & \cdots & x_{p_{max}d}
\end{pmatrix}
\begin{pmatrix}
m_1 \\
m_2 \\
\vdots \\
m_d
\end{pmatrix}
=
\begin{pmatrix}
y_1 \\
y_2 \\
\vdots \\
y_{p_{max}}
\end{pmatrix}
\tag{8}
$$

In order to solve the equation, the following two conditions are set: a) if $X$ is square and non-
singular then Eq. (7) can be simply calculated as $m = X^{-1}y$; b) if $X$ is not square or singular,
Eq. (7) needs to be rearranged and $m$ can be get by finding the minimum of:
$|Xm - y|^2$                                                                                        (9)
with the distinct solution of:
$m = X^{\#}y$                                                                                        (10)
where $X^{\#}$ is the pseudo-inverse matrix of $X$ (Penrose, 1955; Penrose, 1956).

## 3. Results

In this section, different combinations of input data (Table 3) are adopted to examine their
impacts on hydrological soil moisture estimation. XAJ SMD is used as a hydrological soil
moisture state benchmark for the training and testing. More discussion about the misconception
between the hydrological model's soil moisture state variable and the real-world soil moisture
content is covered in Section 4. During GT and $M$-test processes, all data inputs need to be
normalised so that their mean is zero and standard deviation is 0.5. This step is necessary in
reducing the impacts of numerical difference from various inputs, hence improves the GT
efficiency (Remesan et al., 2008). Five statistical indicators are used for the soil moisture
estimation analysis: Pearson product moment correlation coefficient (*r*), *MSE* which is the
same value as the Gamma statistic $\Gamma$, Standard error (*SE*), *NSE* (Nash and Sutcliffe, 1970), and
Root Mean Square Error (*RMSE*).

**3.1 Scheme 1: SMD Estimation Using SAC-SMA-SM as input**

Although in this scheme, there is no need for data feature selection because only one data input
is involved, the GT is still carried out to explore the useful information about the underlying
relationship between the XAJ SMD and the SAC-SMA-SM. The calculated Gamma statistics
are shown in Table 4. The $\Gamma$ of 0.072 indicates that the optimal *MSE* achievable using any
modelling technique is 0.072; and the small value of *SE* means the precision and accuracy of
the GT result. $\Gamma$ is a significant target value in the *M*-test to find the most suitable training data
size. As presented in Figure 4a, when more training data (i.e., *M* increases in steps of one) is
used the $\Gamma$ changes dramatically. Eventually at $M = 292$, $\Gamma$ starts to stabilise around 0.072. The
*M*-test allows us to confidently apply the first 292 datasets to build a model of a given quality,
in the sense of predicting with a *MSE* around the asymptotic level. The corresponding Gamma
gradient (*A*) suggests the complexity of the underlying system: the larger the *A* value is the
more complex the system is. For example if *A* is significantly large, a more complicated model
like a Support Vector Machine might be required, but $A = 1.353$ in Scheme 1 is small (Remesan
et al., 2008), therefore a LLR model should be able to simulate the system. For LLR modelling,
its complexity level is controlled by the $p_{max}$ parameter. As illustrated in Figure 5, $p_{max}$ is
identified from a trial and error method. The procedure is by increasing the LLR $p_{max}$ value
from 2 to 100 to analyse the variations of their corresponding $\Gamma$ results. It can be seen from
Figure 5 that the smallest $\Gamma$ is achieved at $p_{max} = 4$, which is therefore adopted for the LLR
modelling. The training and testing scatter plots for the LLR modelling are shown in Figure 6a.
It is observed that there are some points lying far above the bisector line during the training
period signifies higher estimations whereas some points sit far below the bisector line during
the testing period indicates under-estimation of the SMD. For the testing results, when XAJ
simulated soil moistures state have already reach the total dryness (i.e., XAJ SMD peaks at
around 0.080 m) the predicted soil moisture state is still in the drying progress. Figure 7a plots
the time series of the estimated and the targeted SMD. The plot shows that the estimated SMD
follows the seasonal trend of the soil moisture fluctuations well, so it is wetter during the winter
season and exsiccated during the hot summer season. However it is clear to see that the model
is not able to capture the extreme situations very well, especially during the wet season when
the XAJ SMD becomes smaller (e.g., between Day 300 and Day 350).
**3.2 Scheme 2: SMD Estimation Using SAC-SMA-SM and MODIS-LST as inputs**
Land surface temperature is the product of the soil temperature multiplied by the emissivity,
and the emissivity depends on the dielectric constant of the soil and soil moisture (Rodriguez-
Fernandez et al., 2015). Therefore the additional MODIS-LST information could potentially
improve the soil moisture estimation. The modelling process is the same as in Scheme 1. In
Table 4, it is clear to observe that by adding the MODIS-LST input, the $\Gamma$ is improved to 0.060
and its corresponding gradient $A$ is reduced significantly to less than half of the Scheme 1's.
Meanwhile the $SE$ value is decreased remarkably as well showing the accuracy of the GT. The
$M$-test in Figure 4b shows the graph settles to an asymptote around 0.060 which is consistent
with the calculated $\Gamma$ result. Training data size of 199 is chosen here because it gives the lowest
$\Gamma$ value. For the LLR modelling, the best $p_{max}$ value is found to be 2 from the trial and error
result in Figure 5. The LLR training and testing performances are presented in Figure 6b.
Although the problem of underestimation of extremely dry soil still exists (i.e., the points
concentrate at the right end of the training and testing plots), overall the model's prediction
ability during both phases are better than Scheme 1's (i.e., data points are closer to the 45° line).
The improvement can also be seen clearly in the time series plot in Figure 7b. For example, the
big disparities between the estimated and the targeted SMDs around DAY 300 and DAY 350
are reduced evidently.
**3.3 Scheme 3: SMD Estimation Using SAC-SMA-SM and SMOS-$T_b$s as inputs**
The multi-angle $T_b$s retrievals are the main data inputs for SMOS soil moisture calculation,
therefore their inclusion should also add a positive effect to the hydrological soil moisture
estimation. As aforementioned, an efficient feature selection of the SMOS incidence angles is
important for the best SMD calculation. In this study all the possible combinations from all
inputs variables are examined with the $\Gamma$ result as the statistical indicator. This method is
capable of examining every combination (16383 embeddings in this case) of data inputs to
target the optimal combination that gives the smallest absolute $\Gamma$ value. As discussed in Section
2.3.4, SAC-SMA-SM is a compulsory data input, so it is not included in the selecting process.
The best set of SMOS-$T_b$s to retrieve soil moisture state is composed of H polarization at the
incidence angles of 27.5$^o$-47.5$^o$, 57.5$^o$, and V polarization at the incidence angles of 27.5$^o$-42.5$^o$,
52.5$^o$, 57.5$^o$. This result demonstrates that using a combination of H and V $T_b$s gives a better
soil moisture estimation, which is logically sensible because different polarizations carry
distinct information of the Earth surface. However some incidence angles could held common
features which when putting together could result in a negative effect to the LLR modelling,
and are therefore not included. The detailed investigation of the possible common features is
out of the scope of this paper which is mainly due to the SMOS working mechanism.
As seen from Table 4, the inclusion of SMOS-$T_b$s significantly improves the $\Gamma$ result by 54%,
while the gradient $A$ is reduced greatly by 89% as compared with Scheme 1. The small $A$ value
illustrates that the underlying system is more straightforward and easier to model than the
Scheme 1's. The $M$-test analysis in Figure 4c produces an asymptotic convergence from 120
training data size of $\Gamma$ value around 0.033. It is interesting to see that the proportion of the
required training data is relatively larger than those in Scheme 1 and 2. The potential reason
could be explained by the larger amount of data inputs in this scheme. For LLR modelling, the
$p_{max}$ that gives the smallest $\Gamma$ is 7 (Figure 5). The SMD estimations during the training and the
testing are presented in Figure 8a. It can be seen that the SMD prediction ability of this scheme
is remarkably better than the previous ones, as most of the points lie on the bisector line albeit
there are still some under- and over- estimations. The reason SMOS outperforms MODIS in
SMD estimation could be due to the long wavelength microwave has, so it presents the top few
centimetres of the soil while MODIS LST (thermal infrared) only provides information at the
soil surface. The used LLR algorithm has been double checked to filter out the potential of
overfitting problem. The checking processes are performed by muddling the SMD target in the
testing datasets as well as altering the input file, and its efficiency stays the same. Hence it is
believed that the LLR model is very useful in calculating SMD from this scheme. Generally
the *NSE*, *r* and *RMSE* statistical indicators show a high agreement during both training and
testing phases. For the time series plot in Figure 7c, it is clear to see that most of the estimated
points lie closely to the benchmark line. The observed outliers could be partly due to the data
shortage in this scheme, so that not all the scenarios are covered in the datasets.
**3.4 Scheme 4: SMD Estimation Using SAC-SMA-SM, MODIS-LST, and SMOS-T$_b$s as**

**inputs**

In this scheme, all the three data sources are used to test if the modelling performance can be
further improved. Here the full embedding calculation is again carried out to explore the most
suitable incidence angles from the SMOS-T$_b$s. This is because the added MODIS-LST data
could carry identical (i.e., redundant) features with some of the SMOS-T$_b$s datasets. As a result
of the full embedding calculation, the best set of SMOS-T$_b$s is composed of H polarization at
the incidence angles of 37.5º-57.5º, and V polarization at the incidence angles of 37.5º-42.5º,
57.5º. As seen in Figure 4d, the total amount of data is significantly reduced due to the shortage
of simultanuously available days between the MODIS and the SMOS observations.
Interestingly the *M*-test graph vibrates more significantly than the other three schemes, which
could be due to the smaller data size and the larger amount of data inputs in this scheme. Here
the training data size is chosen as 62 with $\Gamma$ obtained at around 0.030. The optimal $p_{max}$ is
identified to be 5 (Figure 5). The LLR modelling results are shown in Figure 7d and Figure 8b.
It is obvious that this scheme further improves the accuracy of the SMD estimation, especially
with the high statistical performances achieved during both training and testing phases.
Comparatively this scheme is more stable for SMD estimation, albeit it requires more data
inputs and is only realisable when both the MODIS and the SMOS observations are available.

**3.5 Produce an Unintermitted Soil Moisture Product**

The data availability of the four schemes varies. As shown in Figure 9, Scheme 1 which has
the poorest soil moisture state estimation gives the most data availability, while Scheme 4
which has the most accurate soil moisture state estimation owns the least data availability. In
order to produce an unintermitted hydrological soil moisture product, the four schemes need to
be combined together to complement each other. The combining method is by selecting the
best available soil moisture estimation. For example if all the schemes have available data at
the same time, the best scheme's soil moisture data is chosen (i.e., scheme 4 in this situation);
whereas if just one scheme has data on that day, only that scheme's soil moisture data is used.
The performances of the four schemes as well as the combined product are summarised in
Table 5. Although the combined soil moisture state is obtained with lower statistical
performances than Scheme 3's and 4's, it is still hydrologically very accurate especially when
comparing with the SMOS's official soil moisture product (Table 5). The time series of the
combined soil moisture state is plotted in Figure 10. It can be seen that the general trend of the
produced soil moisture state follows the targeted data very well. However it tends to
overestimate some of the wet events during the rainy season and significantly underestimate
the dryer soil condition in September 2011. Those poor estimations are mostly from the Scheme
1 and 2 where Schemes 3 and 4 are not available. Since more and more microwave satellite
observations are becoming obtainable, those new data sources could add extra benefits into the
proposed model, and the accuracy of the soil moisture product is expected to be further
enhanced.
**4. Discussion**
-   *What is a soil moisture state variable?*
This study uses the XAJ's SMD simulation as a target because it is hydrological model directly
produced. However it is argued that models with different parameters values can generate
equally good flow results named as the equifinality effect, because they are all calibrated based
on the observed flow. For this reason, their soil moisture state variables can be distinct among
each other.
In order to investigate this effect in more details, the XAJ model is manipulated by increasing
one of its parameters *WUM* by 30 %. By doing so, the XAJ's flow simulation remains as
effective as its original form (the same *NSE* values), but its soil moisture state changes
significantly from its original values. For a better visualisation, an enlarged plot of the SMD
simulations between Day 222 and Day 344 is presented. As seen from Figure 11a although the
soil moisture state variables from two equally good calibrations have a wide range of value
differences ($NSE = 0.34$), they both follow the same pattern: when it rains they become wet by
the similar amount; when there is a dry period they all move into a dryer state in a similar rate
to the actual evapotranspiration. Therefore they appear as in parallel movements and the latter
plot (Figure 11b) shows a very strong linear correlation ($r = 1.0$) between them. It is important
to note that the selection of the dry period (i.e., high SMD values) is because it is the most
critical period of time for the need of accurate soil moisture values for hydrological modelling.
This is because during the real-time flood forecasting, after a long period of dryness, the
accumulation of error in the hydrological models can become larger and larger with time. With
accurate soil moisture information, the error could be corrected.
Although the absolute values of the models' soil moisture state variables are not quite
meaningful and comparable, their variations are the true reflection of the soil moisture
fluctuations in the real-world. This clarification is a very important concept, because there has
been a wide spread of misunderstanding about the hydrological model's soil moisture state and
its connection with the real-world soil moisture.
-    *Soil moisture state normalisation*
One deficiency of this study is that the generated soil moisture state is based on a hydrological
model's SMD simulation, so it is model parameter dependent. It is desirable to produce a soil
moisture indicator which is independent from model parameters and dimensionless with
variables between 0 and 1. Normalised Hydrological Soil Moisture State (NHSMS) indicators
are produced as presented in Figure 12 (corresponding to the SMD simulations shown in Figure
11). The normalisation method is by adopting the following equation:
$$NHSMS = \frac{SMD - \min(SMD)}{\max(SMD) - \min(SMD)} \qquad (11)$$
Such an approach is very effective as demonstrated by the almost identical SMD curves
between the two XAJ simulations. In the future it is planned to use the same process on other
hydrological models to test if the normalised soil moisture indicators are not only model
parameter independent but also model structure independent. Since all hydrological models are
driven by the same physics laws on the conservation of mass, their normalised soil moisture
indicators should respond in a similar way (soil becomes wetter when it rains and drier when
there is no rain). If this is true a new soil moisture product based on NHSMS could be generated
as a routine product by the operational organisations such as NASA and ESA. Such a soil
moisture product will also be very useful to the meteorological and hydro-meteorological fields
in their land surface modelling because the current land surface models suffer from poor
performance in their runoff estimations. As aforementioned, all current soil moisture products
such as those from ESA and NASA are not optimised for different application fields. Our study
gives an example of simulating the soil moisture data targeted to serve the hydrological
community. It is possible other products serving farmers in agriculture, ecologists in the
environment, and geotechnical engineers in construction could be produced using the proposed
method.
-  *Application of the produced soil moisture data*
Another area needs further work is the hydrological application of the produced data. Generally
effective hydrological application of soil moisture data needs three pre-conditions: 1) a good
soil moisture data relevant to hydrology; 2) a hydrological model compatible with such data;
3) an effective data assimilation scheme. This paper tackles the first point, and the other two
points would need further research because there are significant knowledge gaps in them. If all
the three points are solved, such a data has a huge potential in operational hydrological
modelling. For example, initialisation of the model could be shortened which reduces the need
for model warm up. This is important during real-time flood forecasting when there is not
enough data to warm up the model for an imminent flood event. Such a warm-up period could
be very long, as demonstrated by the study in (Ceola et al., 2015). In addition the XAJ SMD
data used here is based on the calibration of the observed rainfall and flow, so that the targeted
SMD is interpolated between observations and there is a minimum time-drift. In the real-time
flood forecasting the errors in precipitation and evapotranspiration could accumulate which
cause time-drift problems. Therefore a soil moisture product such as the one produced in this
study (i.e., based on minimal time-drift SMD) could help avoiding such a problem. The
proposed soil moisture data is also valuable for the validation of land surface models, especially
useful for their runoff simulations. Due to the limit of time and resources this study has not
tackled all the issues, but has laid a good foundation for their future researches.
-   *XAJ model under frozen conditions*
The Pontiac catchment is characterized by soil freezing events in winter seasons. During
freezing events, soil moisture transfer fundamentally differs from the unfrozen conditions (e.g.,
(Gelfan, 2006)). Although the XAJ model has been successfully applied in simulating flows in
frozen soil conditions (e.g., see Zhou et al., 2008) as well as in this case study, the lumped XAJ
model does not explicitly consider soil freezing, thus SMD simulations can be inaccurate for
winter seasons and further research is needed to investigate this issue further.
**5. Conclusions**
A hydrological soil moisture product is produced for the Pontiac catchment using the GT and
the LLR modelling techniques based on four data-input schemes. Three data sources are
considered including the soil moisture product from the SAC-SMA model, the land surface
temperature retrieved by the MODIS satellite, and the multi-angle brightness temperatures
acquired from the SMOS satellite. The four data-input schemes are built from the four
combinations of the data sources. The generated soil moisture product (unintermitted with no
missing data) for a period of two years (2010-2011) is compared with the XAJ hydrological
model's SMD simulation to test its hydrological accuracy. It is concluded that the GT and the
LLR modelling techniques together with the chosen data inputs can be used with high
confidence to estimate an unintermitted hydrological soil moisture product, and the proposed
method could be easily applied to other catchments and fields.
In this study it has been found that different data sources have their own unique information
contents, so that they can complement each other using data fusion technique. Their synergy
can be best achieved to produce an enhanced soil moisture product. In data fusion an important
principle is MRmr (Maximum Relevance minimum redundancy). The soil moisture state in
this study is generated from a large number of data inputs, and their selection is carried out by
the GT which is one of the methods in MRmr. This is the first time that the GT is used in a data
fusion of satellite multiple $T_b$s scans, land surface temperature and external soil moisture
information for producing a hydrological soil moisture product. Future studies should explore
other MRmr methods in addition to GT, to compare if they are more effective input selection
methods. As to the data fusion regression model, LLR is chosen in this study because it is easily
applied and very effective. However it is possible there may exist other better models. We
encourage the community to apply the proposed methodology using other regression models.
**Acknowledgments**
This study is supported by Resilient Economy and Society by Integrated SysTems modelling
(RESIST), Newton Fund via Natural Environment Research Council (NERC) and Economic
and Social Research Council (ESRC) (NE/N012143/1). We acknowledge the U.S. Geological
Survey for making available daily streamflow records (http://waterdata.usgs.gov/nwis/rt). The
NLDAS-2    data    sets    used    in    this    article    can    be    obtained    through    the
http://ldas.gsfc.nasa.gov/nldas/NLDAS2forcing.php website, the SMOS Level-3 brightness
temperatures and soil moisture are from the CATDS at http://www.catds.fr/, and the MODIS
Level-3 land surface temperature can be obtained from the LP DAAC website at
https://lpdaac.usgs.gov/dataset_discovery/modis/modis_products_table/mod11c1.

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

820

**Table 1.** The XAJ model parameters used in the Pontiac catchment.

| Symbol | Model parameters | Unit | Range |
|---|---|---|---|
| $K$ | Ratio of evapotranspiration | [-] | 0.10-1.20 |
| $WUM$ | The areal mean field capacity of the upper layer | mm | 30-50 |
| $WLM$ | The areal mean field capacity of the lower layer | mm | 20-150 |
| $WDM$ | The areal mean field capacity of the deep layer | mm | 30-400 |
| $IMP$ | Percentage of impervious and saturated areas in the catchment | % | 0.00-0.10 |
| $B$ | Exponential parameter with a single parabolic curve, which represents the non-uniformity of the spatial distribution of the soil moisture storage capacity over the catchment | [-] | 0.10-0.90 |
| $C$ | Coefficient of the deep layer that depends on the proportion of the catchment area covered by vegetation with deep roots | [-] | 0.10-0.70 |
| $SM$ | Areal mean free water capacity, which represents the maximum possible deficit of free water storage | mm | 10-50 |
| $KG$ | Outflow coefficient of the free water storage to groundwater relationships | [-] | 0.10-0.70 |
| $KSS$ | Outflow coefficient of the free water storage to interflow relationships | [-] | 0.10-0.70 |
| $EX$ | Exponent of the free water capacity curve | [-] | 1.10-2.00 |
| $KKG$ | Recession constant of the groundwater storage | [-] | 0.01-0.99 |
| $KKSS$ | Recession constant of the lower interflow storage | [-] | 0.01-0.99 |
| $CS$ | Recession constant in the lag and route method for routing through the channel system with each sub-catchment | [-] | 0.10-0.70 |
| $L$ | Lag in time | [-] | 0.00-6.00 |
| $V$ | Parameter of the Muskingum method | m/s | 0.40-1.20 |
| $dX$ | Parameter of the Muskingum method | [-] | 0.00-0.40 |

**Table 2.** General data-input properties relevant for this study.

| | SMOS-$T_b$s | MODIS-LST | SAC-SMA-SM |
|---|---|---|---|
| Product | brightness temperature | land surface temperature | soil moisture |
| Unit | Kelvin (K) | Kelvin (K) | $m^3/m^3$ |
| Near-Real-Time (NRT) | Yes | Yes | Yes |
| Spatial resolution (km) | 35-50 | 5.6 | 14 |
| Data time-step | ~ every three days | ~ daily | Daily |
| Data availability for the studying period (days) | 217 | 458 | 730 |

**Table 3.** Four data-input schemes: scheme 1: SAC-SMA-SM; scheme 2: SAC-SMA-SM and MODIS-LST; scheme 3: SAC-SMA-SM and SMOS-$T_b$s; scheme 4: SAC-SMA-SM, MODIS-LST, and SMOS-$T_b$s.

|          | SAC-SMA-SM | MODIS-LST | SMOS-$T_b$s |
|----------|:----------:|:---------:|:-----------:|
| Scheme 1 | x          |           |             |
| Scheme 2 | x          | x         |             |
| Scheme 3 | x          |           | x           |
| Scheme 4 | x          | x         | x           |

**Table 4.** Model statistical performances and modelling information, where $\Gamma$ is the calculated gamma statistic which is the minimum *MSE* that can be achieved from a modelling method; *A* is the Gamma gradient; *SE* is the Standard error; $p_{max}$ is the nearest points for LLR modelling; *M* is the training data size; and SMOS IA is the chosen incidence angles of SMOS-T$_b$s.

| | $\Gamma$ | $A$ | $SE$ | $p_{max}$ | $M$ | SMOS IA |
|---|---|---|---|---|---|---|
| Scheme 1 | 0.072 | 1.353 | 0.004 | 4 | 292 | - |
| Scheme 2 | 0.060 | 0.568 | 0.002 | 2 | 199 | - |
| Scheme 3 | 0.033 | 0.152 | 0.004 | 7 | 120 | H: 27.5°-47.5°, 57.5° |
| | | | | | | V: 27.5°-42.5°, 52.5°, 57.5° |
| Scheme 4 | 0.029 | 0.119 | 0.006 | 5 | 62 | H: 37.5°-57.5° |
| | | | | | | V: 37.5°-42.5°, 57.5° |

**Table 5**. Summary of SMD estimation performances. It is noted that *RMSE* is in the unit of metre.

| | Training | | | Testing | | |
|---|---|---|---|---|---|---|
| | *NSE* | *r* | *RMSE* | *NSE* | *r* | *RMSE* |
| Scheme 1 | 0.752 | 0.870 | 0.011 | 0.688 | 0.830 | 0.014 |
| Scheme 2 | 0.767 | 0.877 | 0.011 | 0.747 | 0.865 | 0.012 |
| Scheme 3 | 0.928 | 0.965 | 0.006 | 0.876 | 0.940 | 0.008 |
| Scheme 4 | 0.912 | 0.957 | 0.007 | 0.912 | 0.960 | 0.007 |
| Combined | - | - | - | 0.790 | 0.889 | 0.011 |
| SMOS-SM | - | - | - | 0.420 | 0.650 | 0.017 |

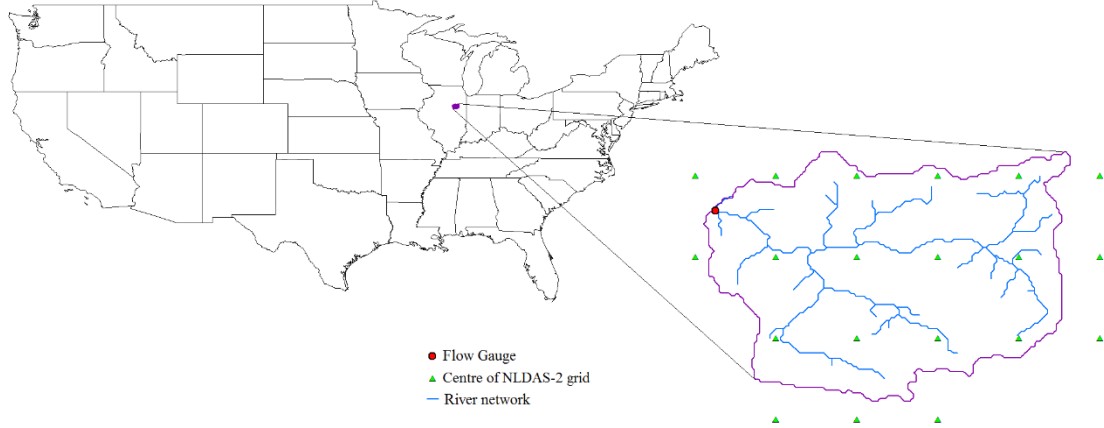

**Figure. 1.** The location and river network of the Pontiac catchment in the U.S., with the flow gauge and NLDAS-2 central grid points (Zhuo et al., 2015a).

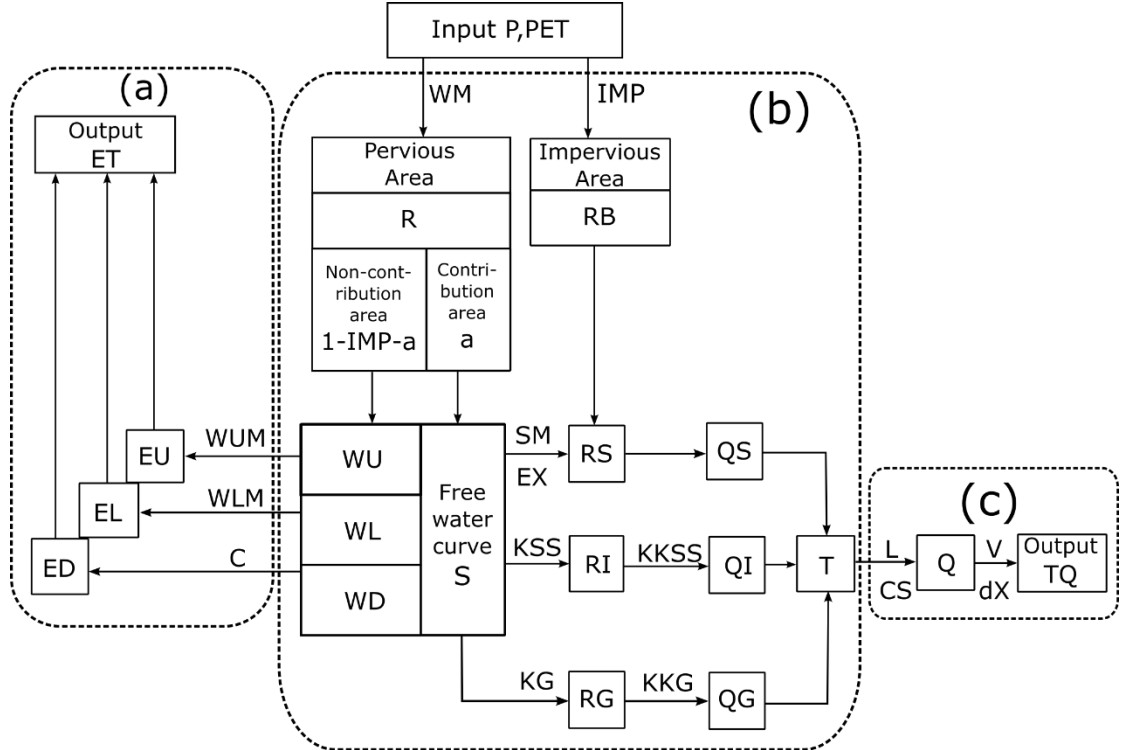

**Figure. 2.** Adopted flowchart of the XAJ model *(Zhao, 1992)*. The model consists of an evapotranspiration component (a), a runoff generating component (b), and a runoff routing component (c). *P*, *PET*, and *ET* are the precipitation, potential evapotranspiration, and the simulated actual evapotranspiration respectively; *WU*, *WL* and *WD* represent the upper, lower, and deep soil layers' areal mean tension water storage respectively; *WM* is the areal mean field capacity; *EU*, *EL,* and *ED* stand for the upper, lower, and deep soil layers' evapotranspiration output respectively; *S* is the areal mean free water storage; *a* is the portion of the sub-catchment producing runoff; *IMP* is the factor of impervious area in a catchment; *RB* is the direct runoff produced from the small portion of impervious area; *R* is the total runoff generated from the model with surface runoff (*RS*), interflow (*RI*), and groundwater runoff (*RG*) components respectively. These three runoff components are then transferred into *QS*, *QI,* and *QG* and combined as the total sub-catchment inflow (*T*) to the channel network. The flow outputs *Q* from each sub-catchment are then routed to the catchment outlet to produce the final flow result (*TQ*). The rest of the symbols are explained in Table 1.

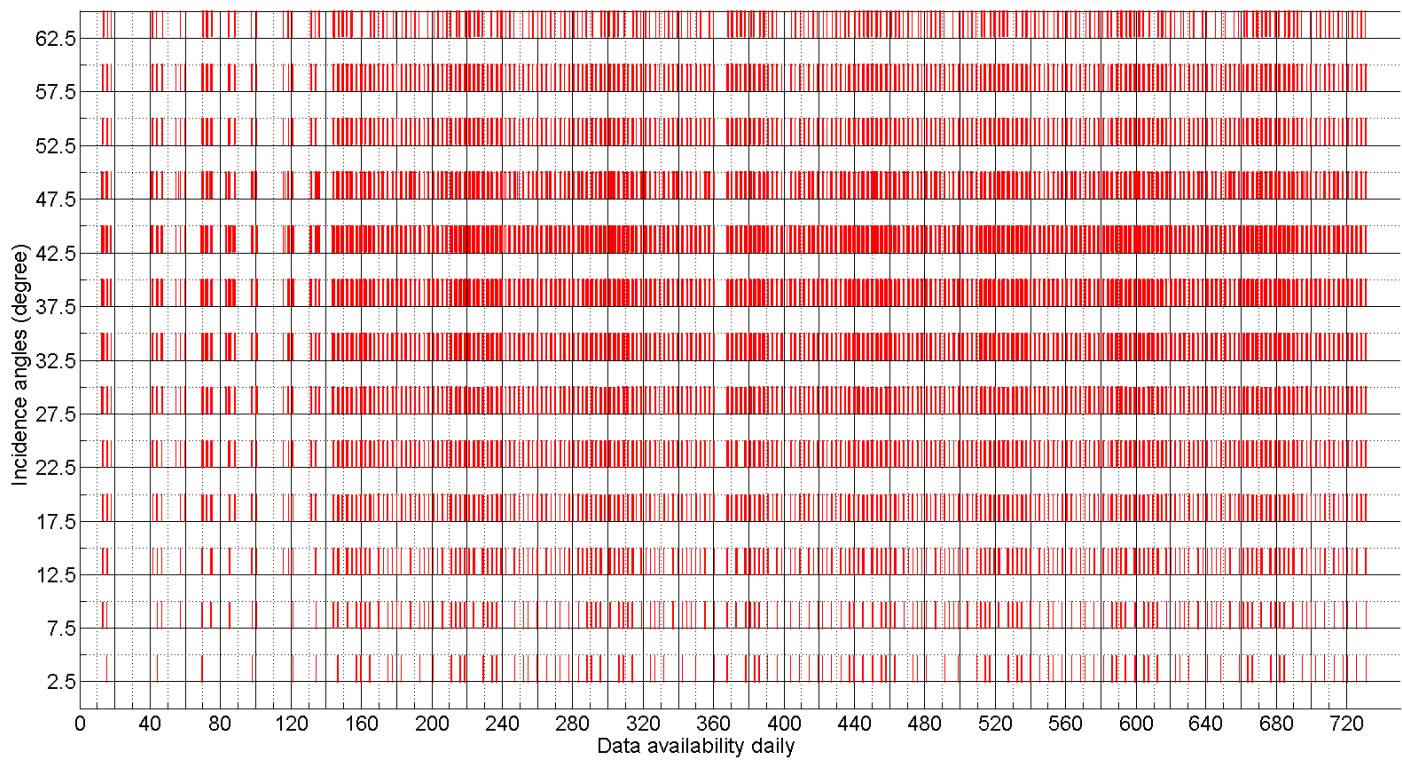

**Figure 3.** SMOS-$T_b$s data availabilities. It is noted that the available dates for the horizontal and the vertical polarizations are the same, so only one is shown here.

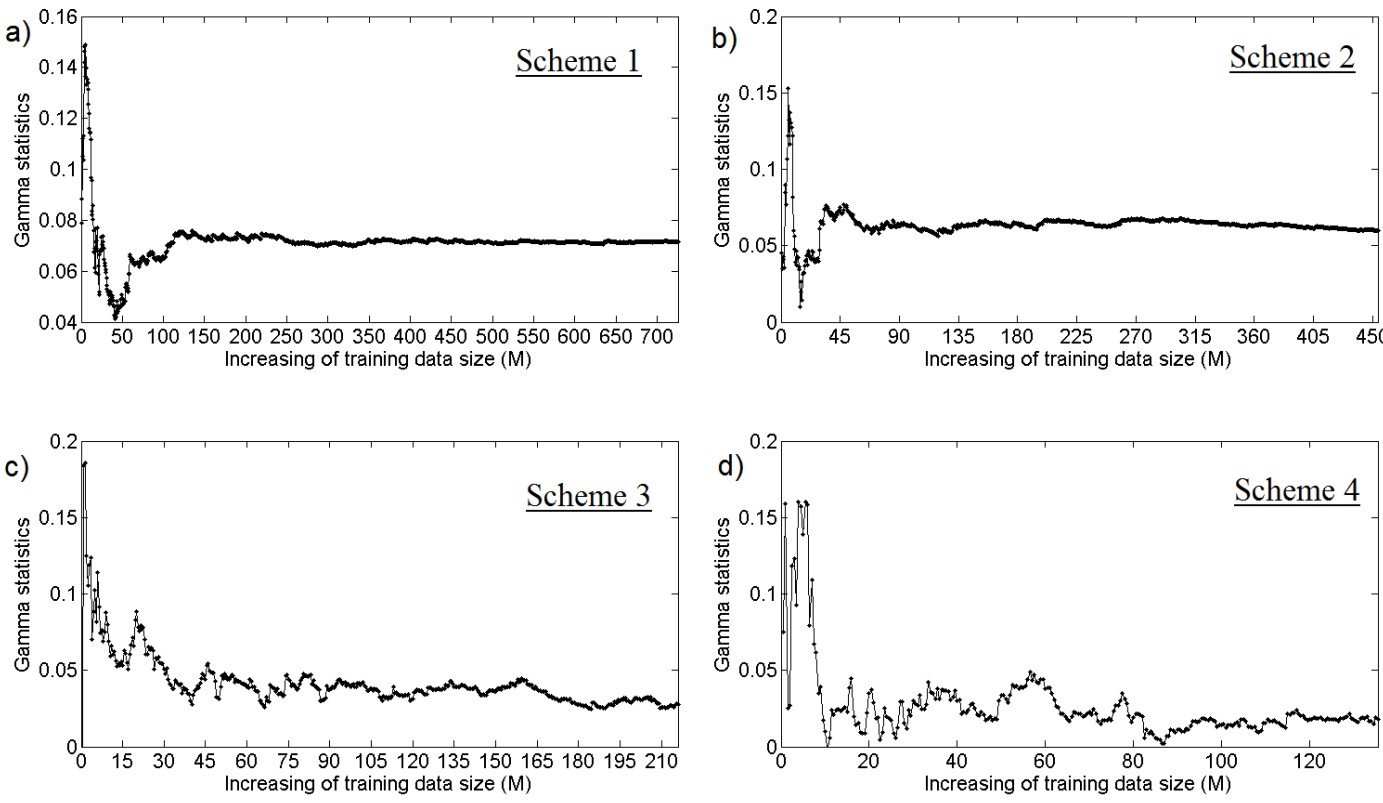

**Figure 4.** *M*-test, to find the best training data size: a) Scheme 1; b) Scheme 2; c) Scheme 3; and d) Scheme 4.

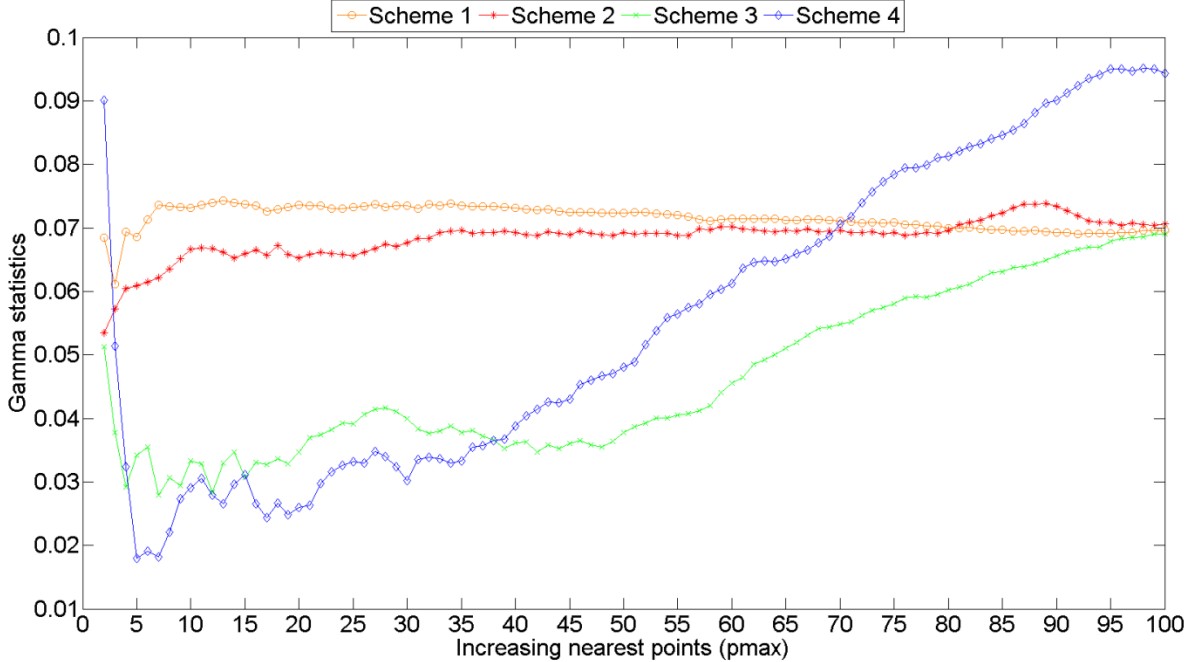

**Figure 5.** Gamma statistic ($\Gamma$) variations for increasing the LLR $p_{max}$ value.

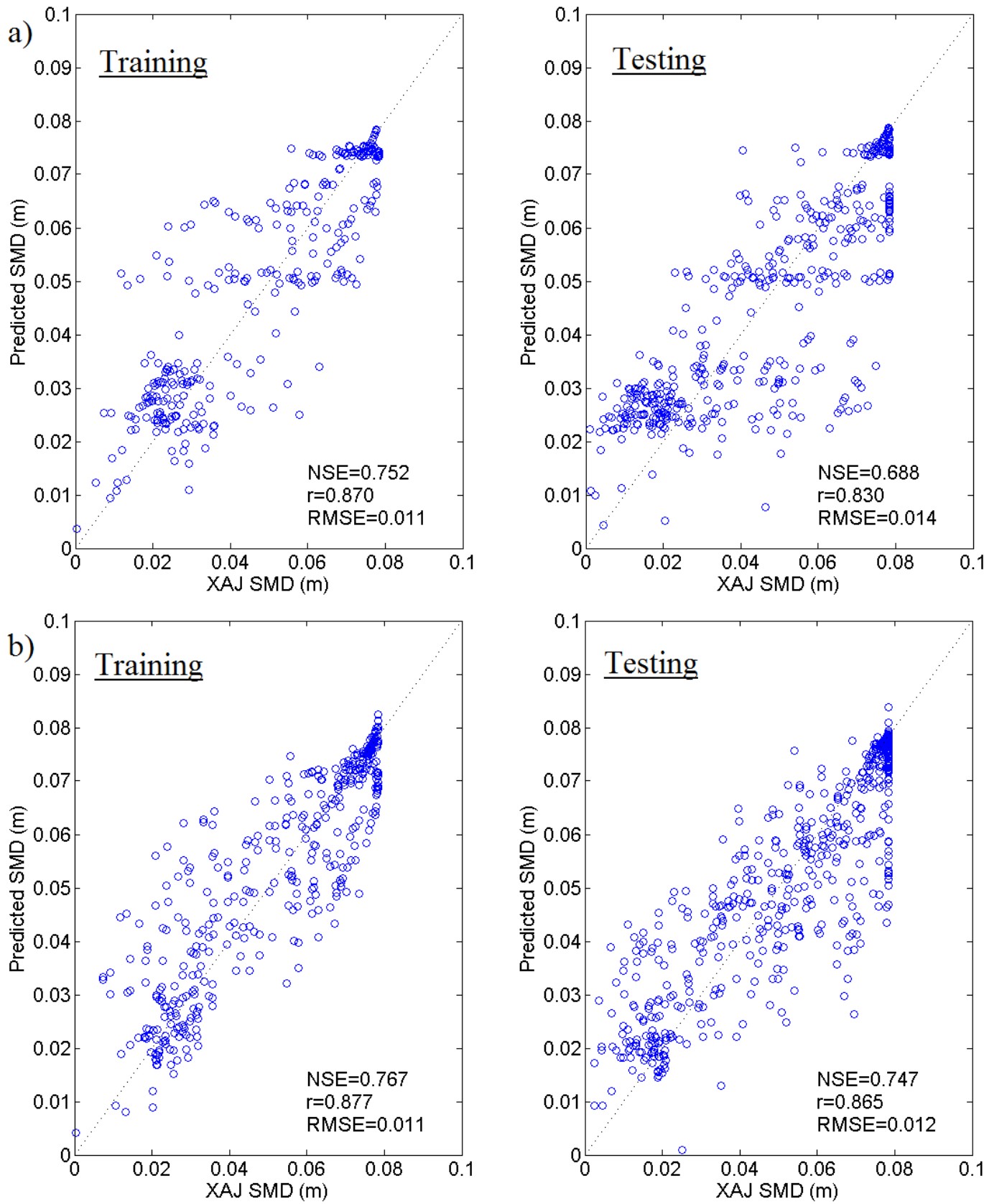

**Figure 6.** LLR modelling during the training and testing phases for a) Schemes 1 and b) Scheme 2.

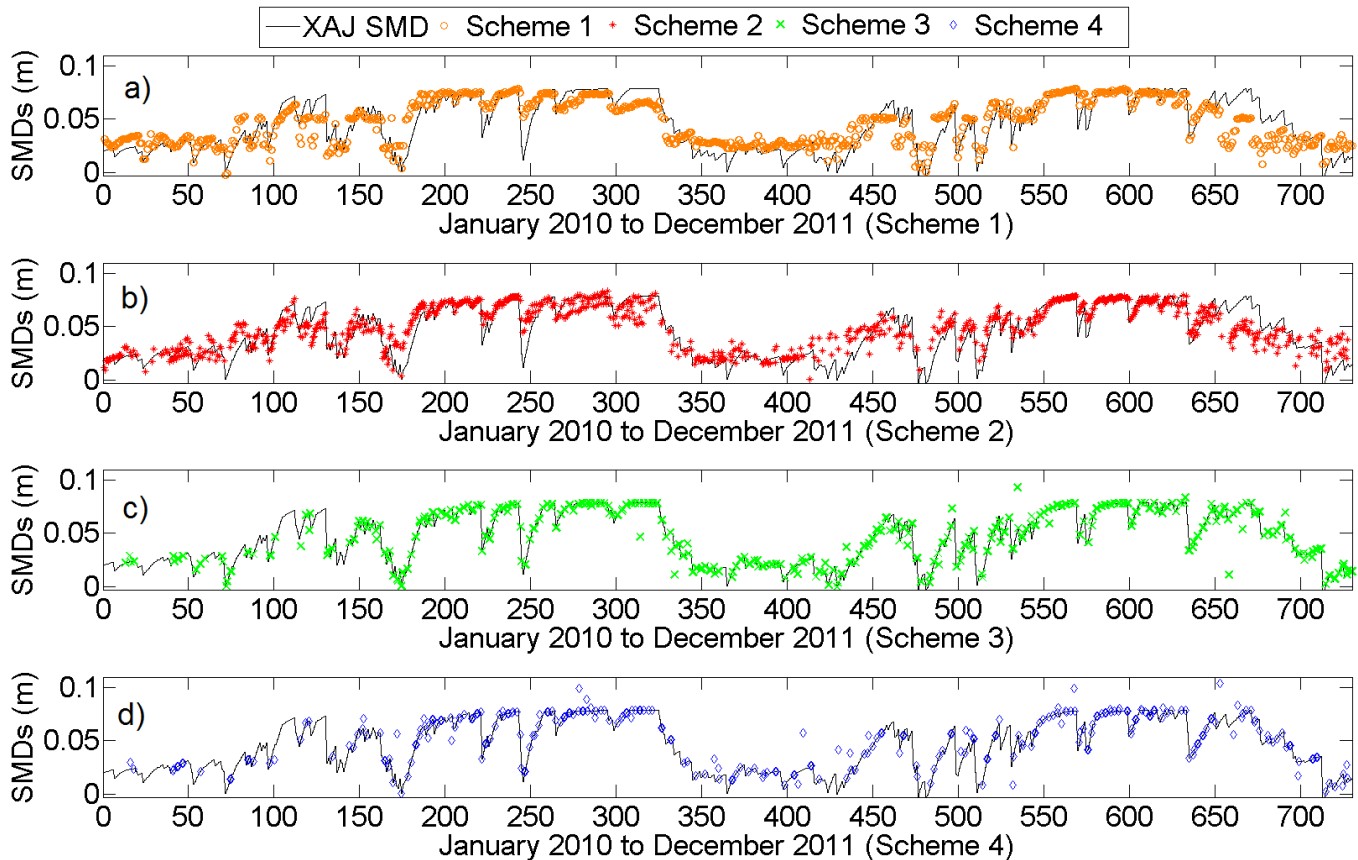

**Figure 7.** The time series plots of the XAJ SMD and the estimated SMD from the four schemes: a) Scheme 1; b) Scheme 2; c) Scheme 3; and d) Scheme 4.

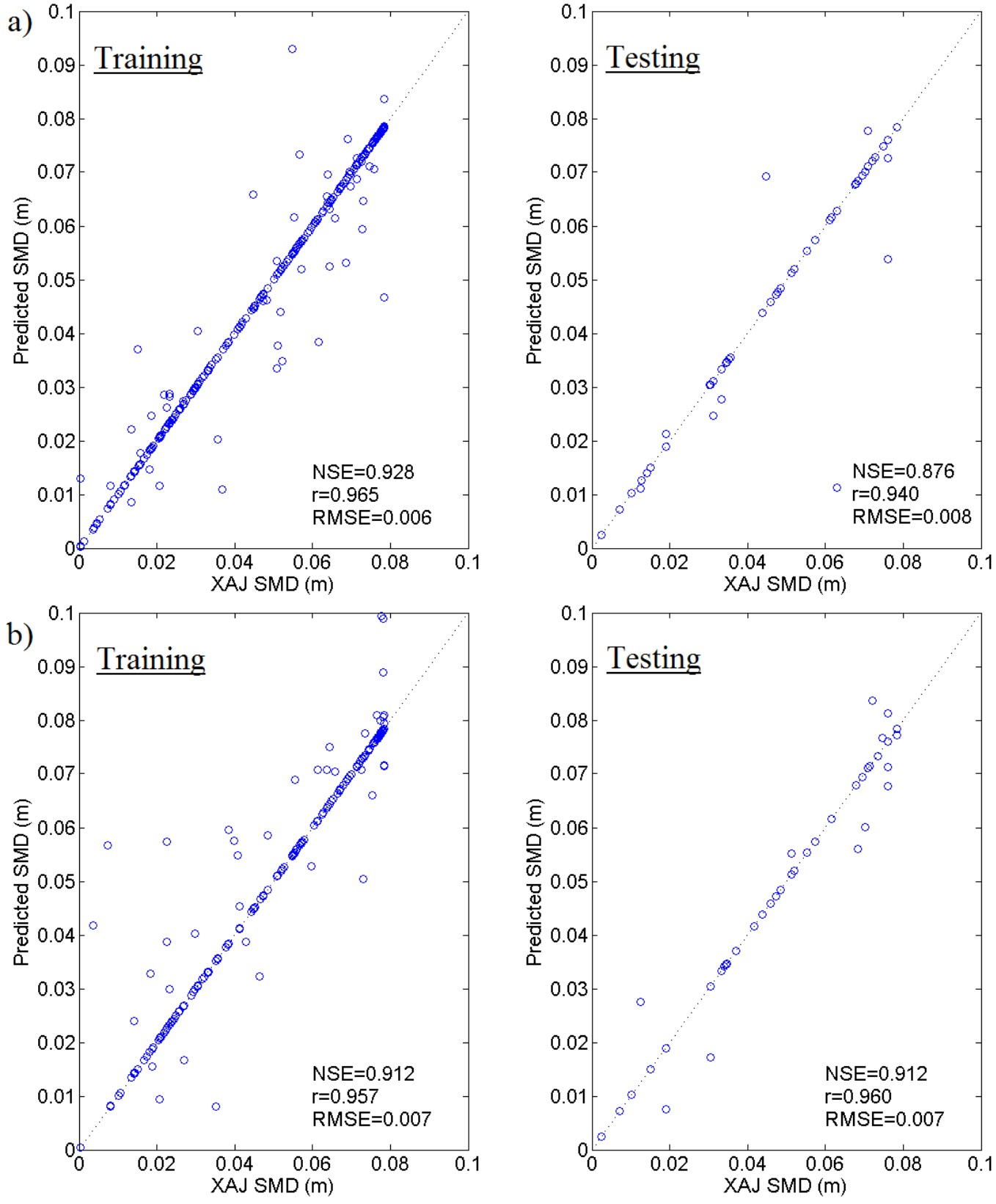

**Figure 8.** LLR modelling during the training and testing phases for a) Schemes 3 and b) Scheme 4.

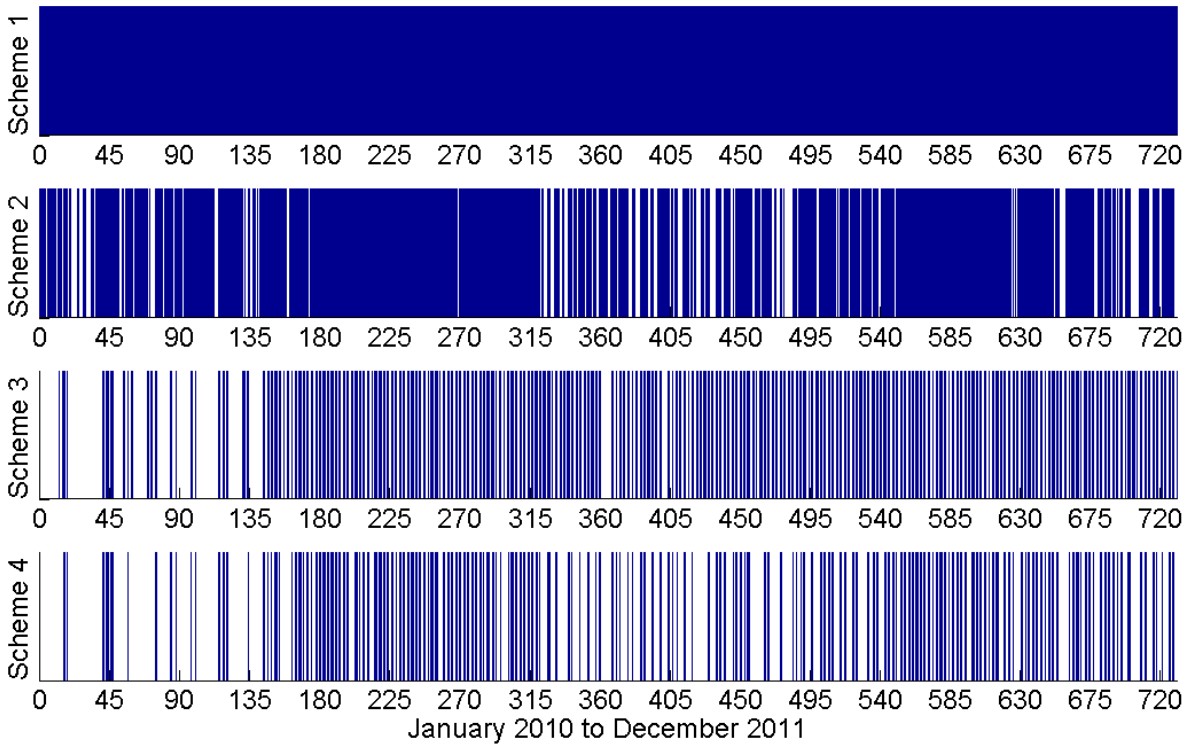

**Figure 9.** Data availability plots of the four schemes: Scheme 1: SAC-SMA-SM input; Scheme 2: SAC-SMA-SM and MODIS-LST inputs; Scheme 3: SAC-SMA-SM and SMOS-$T_b$s inputs; Scheme 4: SAC-SMA-SM, MODIS-LST, and SMOS-$T_b$s inputs. The total available days for the four schemes are 730, 458, 217, and 140 respectively.

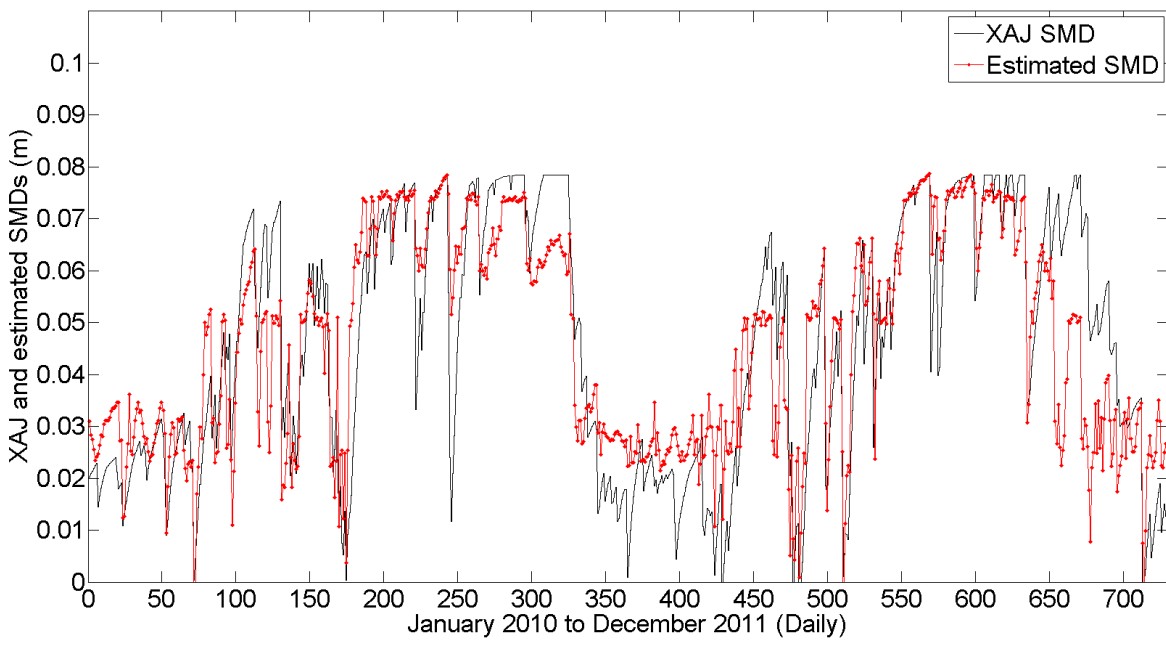

**Figure 10.** Time series plot of the combined daily hydrological soil moisture state estimations.

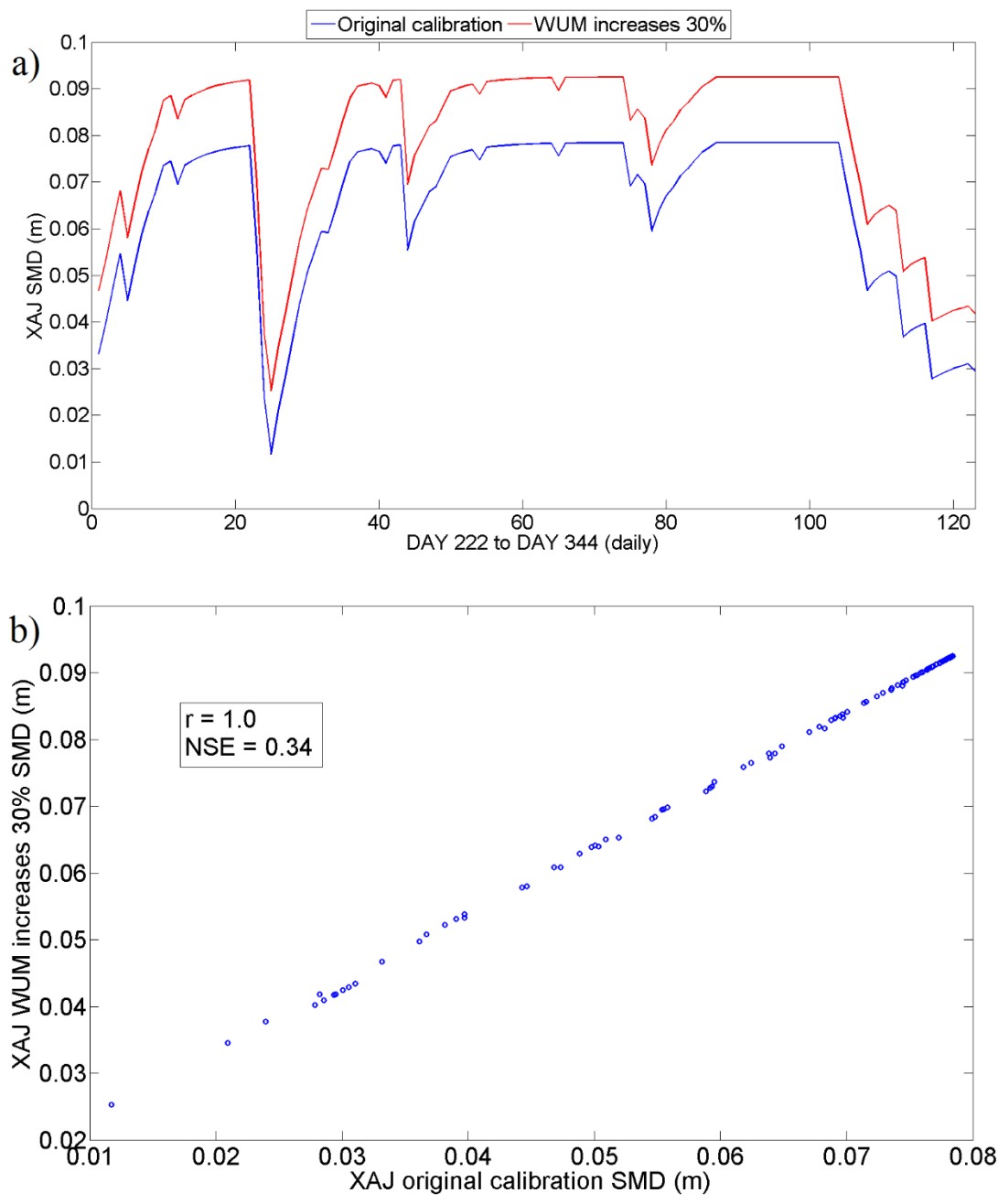

**Figure 11.** SMD variations from the manipulated XAJ calibration (i.e., the WUM parameter is increased by 30 %) and its original calibration.

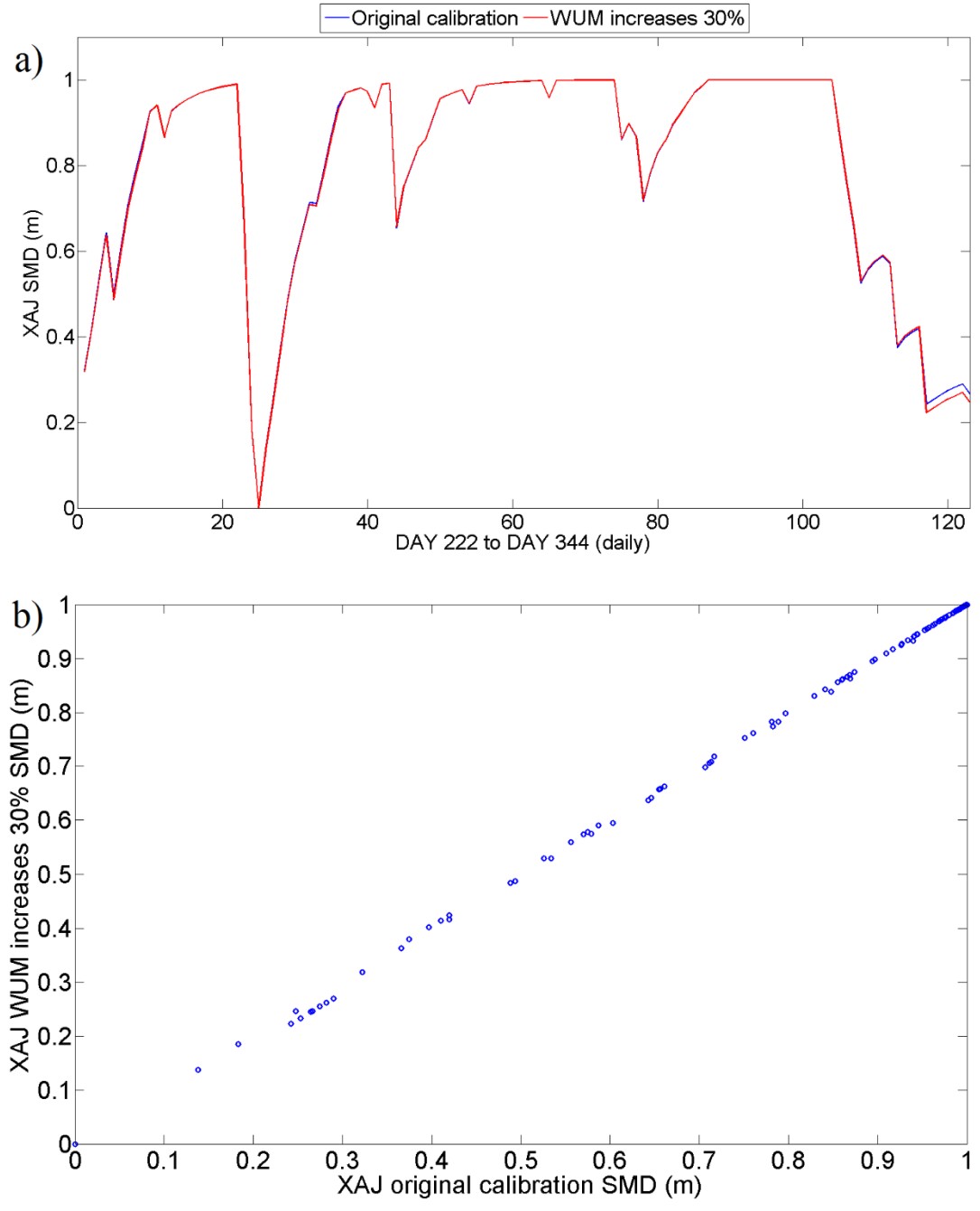

**Figure 12.** Normalised SMD variations from the manipulated XAJ calibration (i.e., the WUM parameter is increased by 30 %) and its original calibration.