# Peer review of "Multi-source hydrological soil moisture state estimation using data fusion optimisation"

_Hydrology and Earth System Sciences, 2016_

## Referee Comment (RC1) · Anonymous Referee #1 · 14 Feb 2017

Referee Overview on the paper by Lu Zhuo and Dawei Han "Multi-source hydrological soil moisture state estimation using data fusion optimization" published for discussion in Hydrol. Earth Syst. Sci. Discuss., doi:10.5194/hess-2016-478, 2016

General comments. The paper is devoted to investigation of ability to estimate land surface soil moisture with the help of the Local Linear Regression model using data retrieved from hydrological model SAC-SMA (soil moisture), radiometer MODIS (land surface temperature), and SMOS (brightness temperatures in H-V polarizations). The research topic is very relevant. The described approach is original and the results is quite convincing.

Specific comments. These comments concern with literature redaction of the paper. May be the Introduction is too long and its text is partly repeated in different sections

of the paper. The text is quite loose and without prejudice to the content may be compressed in different parts of the paper especially in the beginnings of the sections.

Technical corrections 1. It is necessary to correct "polarisation" for "polarization" (for instance, line 13 and further). This mistake can be corrected using contextual replacement.

There are some faults in the References.

Line 602. Is winGammaTM the beginning of PhD Thesis title?

Line 642. smos mission has be replaced by SMOS mission

Lines 656-657. JOURNAL OF GEOPHYSICAL RESEARCH-ALL SERIES-, 99: 14,415-14,415. The letters must be in lower case. And what about pages?

Line 701. Number of pages is not specified

Lines 780-783 One paper is mentioned instead of two different ones.

780 Zhao, R.-J., 1992a. The Xinanjiang model applied in China. Journal of Hydrology, 135(1): 781 371-381. 782 Zhao, RenJun, 1992b. The Xinanjiang model applied in China. Journal of Hydrology, 135(1): 783 371-381

Resume The submitted paper is interesting; it corresponds to the subject of the journal. The paper may be published in the HESS after mentioned corrections.

---

## Author Comment (AC1) · 14 Feb 2017

Replies to Referee 1

Reply: Agreed. The Introduction will be condensed in the revised manuscript, and the technical faults will also be amended as suggested.

---

## Referee Comment (RC3) · Anonymous Referee #2 · 26 Mar 2017

The authors have suggested a method for creating a soil moisture product for hydrological applications using multiple data sources retrieved from three sources (SAC-SMA land surface modelling product, MODIS satellite-retrieved land-surface temperature, and Soil Moisture and Ocean Salinity (SMOS) project data) using the Gamma-test and the Local Linear Regression techniques. The accuracy of the produced soil moisture data was evaluated against the Xinanjiang (XAJ) hydrological model's soil moisture simulations. The authors have concluded that "together with the chosen data inputs can be used with high confidence to estimate an unintermitted hydrological soil moisture product, and the proposed method could be easily applied to other catchments and fields". The topic is of current scientific interest and the manuscript is overall well prepared. However, there are some general points that need to be clarified and at some points more detailed information or analysis is necessary. The following general and specific comments should be addressed before accepting this manuscript for publication.

General Comments

1. Duration of the study period is two years: from January 2010 to December 2011. In my opinion, the presented calibration/validation results don't allow a reader to evaluate the model applicability and the aforementioned conclusion looks too optimistic. The point is that the study period is too short, and the presented results of the model validation are deficient. It means that the overall model performance based on these results is very sensitive to the meteorological conditions of the study period and the performance assessments are rather casual. This fact limits opportunities for application of the proposed method "to other catchments and fields". The conclusion on the model applicability would be more convincing if the authors evaluated the model against hydrological data for a longer period. According to the USGS Water Resources webpage, streamflow data series exceed 10 years for the Vermilion River at Pontiac, IL.

2. The Pontiac catchment is characterized by frequent soil freezing events in winter seasons. During freezing events, soil moisture transfer fundamentally differs from the unfrozen conditions (e.g. Gelfan, 2006). To my knowledge, the lumped XAJ model does not consider soil freezing, thus SMD simulations can be inaccurate for winter seasons. Please clarify. Gelfan A. N. (2006) Physically based model of heat and water transfer in frozen soil and its parametrization by basic soil data. IAHS Publ., 303, pp. 293-304.

3. The authors argued that "only the surface SMD referring to the vegetation and the very thin topsoil, is utilised as a hydrological soil moisture target". Does the XAJ model allow one to simulate SMD in the "very thin topsoil"? If no, this should be clearly pointed out in the manuscript, and the simulation results' interpretation should be corrected.

4. I fully agree with the authors that the results are "model parameter dependent" (line 486). But I disagree that the proposed NHSMS indicators allow one to obtain

independent results. I think that Figure 13 can not be considered as an evidence of such independence because of at least two reasons: (1) only one parameter has been changed; (2) the obtained closeness of the two curves is shown for only 4 months (of 2 years) with mostly high SMD values. Thus I believe that the results presented in the manuscript are dependent on the XAJ model structure, parameters and inputs. Please give a comment.

5. Conclusion, lines 539-541. I do not share the optimistic view of the authors on the perspective of the proposed fusion technique. Yes, the data sources contain part of useful (and probably independent) information. However, these data contain their own large measurement errors and error's synergy can result in dramatically increase of the presented results' uncertainty. I would like to read the authors' comment on this topic.

Specific Comments 1. Lines 133-136: The phrase beginning from the words "It is worth noting that..." looks unnecessary in scientific (non-popular) hydrological text 2. I suggest removing Fig. 3. This figure has been already demonstrated in three (at least) recently published papers of the authors (Zhuo and Han, 2016, 2017; Zhuo et al., 2015) 3. Line 165: There are 17 parameters in Table 1 4. Eq. 1: M is not defined 5. Lines 495-496: The statement "all hydrological models are driven by the same hydrological inputs (precipitation, evapotranspiration and flow)" is misconception. Please be more precise.

---

## Author Response (AR1)

**To Editor**

Dear Prof. Gelfan,

Thank you very much for accepting the paper with minor revision. The manuscript has been amended to address the issues raised in the reviews. The point-by-point replies to the referees' comments are listed below.

If there are any further questions, please let us know.

Lu Zhuo

The Corresponding author (lu.zhuo@bristol.ac.uk )

University of Bristol

**Replies to Referee #1**

...........................................................
General comments. The paper is devoted to investigation of ability to estimate land surface soil moisture with the help of the Local Linear Regression model using data retrieved from hydrological model SAC-SMA (soil moisture), radiometer MODIS (land surface temperature), and SMOS (brightness temperatures in H-V polarizations). The research topic is very relevant. The described approach is original and the results is quite convincing.

Specific comments. These comments concern with literature redaction of the paper. May be the Introduction is too long and its text is partly repeated in different sections of the paper. The text is quite loose and without prejudice to the content may be compressed in different parts of the paper especially in the beginnings of the sections.

Reply: Agreed. Some contents have been moved to Section 2 Material and Methods, so the revised Introduction is now more condensed.

Technical corrections 1. It is necessary to correct "polarisation" for "polarization" (for instance, line 13 and further). This mistake can be corrected using contextual replacement.

Reply: They have been corrected.

There are some faults in the References.

Line 602. Is winGammaTM the beginning of PhD Thesis title?

Reply: Yes. It is the beginning of the PhD Thesis title.

Line 642. smos mission has be replaced by SMOS mission

Reply: It has been corrected.

Lines 656-657. JOURNAL OF GEOPHYSICAL RESEARCH-ALL SERIES-, 99: 14,415-14,415. The letters must be in lower case. And what about pages?

Reply: The reference has been updated.

Line 701. Number of pages is not specified

Reply: The page number has been added.

Lines 780-783 One paper is mentioned instead of two different ones. 780 Zhao, R.-J., 1992a. The Xinanjiang model applied in China. Journal of Hydrology, 135(1): 781 371-381. 782 Zhao, RenJun, 1992b. The Xinanjiang model applied in China. Journal of Hydrology, 135(1): 783 371-381

Reply: They have now been combined as one reference.

Resume The submitted paper is interesting; it corresponds to the subject of the journal. The paper may be published in the HESS after mentioned corrections.

.........................................................

**Replies to Referee #2**

The authors have suggested a method for creating a soil moisture product for hydrological applications using multiple data sources retrieved from three sources (SAC-SMA land surface modelling product, MODIS satellite-retrieved land-surface temperature, and Soil Moisture and Ocean Salinity (SMOS) project data) using the Gamma-test and the Local Linear Regression techniques. The accuracy of the produced soil moisture data was evaluated against the Xinanjiang (XAJ) hydrological model's soil moisture simulations. The authors have concluded that "together with the chosen data inputs can be used with high confidence to estimate an unintermitted hydrological soil moisture product, and the proposed method could be easily applied to other catchments and fields". The topic is of current scientific interest and the manuscript is overall well prepared. However, there are some general points that need to be clarified and at some points more detailed information or analysis is necessary. The following general and specific comments should be addressed before accepting this manuscript for publication.

General Comments

1. Duration of the study period is two years: from January 2010 to December 2011. In my opinion, the presented calibration/validation results don't allow a reader to evaluate the model applicability and the aforementioned conclusion looks too optimistic. The point is that the study period is too short, and the presented results of the model validation are deficient. It means that the overall model performance based on these results is very sensitive to the meteorological conditions of the study period and the performance assessments are rather casual. This fact limits opportunities for application of the proposed method "to other catchments and fields". The conclusion on the model applicability would be more convincing if the authors evaluated the model against hydrological data for a longer period. According to the USGS Water Resources webpage, streamflow data series exceed 10 years for the Vermilion River at Pontiac, IL.

Reply: We agree with the referee's suggestion on using a longer period for model evaluation. However this study has been constrained due to the following two reasons: first although the selected catchment has flow data over 10 years, most of them are discontinuous (e.g., frequent data gaps), and the selected period provides the most complete flow observations. This is essential for the XAJ model's calibration and validation (it is a continuous simulation model instead of an event-based model); second the SMOS satellite was launched in late 2009, so its data products are only available since then. In addition, since this study is a proof of concept for a new method in optimal fusing of multi-source data for soil moisture estimation, it is more effective to inform the hydrological community about such an approach and trial it in a variety of locations than testing it at one location for a few more years.

2. The Pontiac catchment is characterized by frequent soil freezing events in winter seasons. During freezing events, soil moisture transfer fundamentally differs from the unfrozen conditions (e.g. Gelfan, 2006). To my knowledge, the lumped XAJ model does not consider soil freezing, thus SMD simulations can be inaccurate for winter seasons. Please clarify.

Gelfan A. N. (2006) Physically based model of heat and water transfer in frozen soil and its parametrization by basic soil data. IAHS Publ., 303, pp. 293-304.

Reply: The lumped XAJ model has been frequently used in frozen soil conditions (e.g., see Application of Xin'anjiang model in severe cold region of Niqiu River, 2008 (Zhou et al., 2008)). The XAJ model's evapotranspiration component plays a vital role in the model's flow generation. The component constitutes three soil storages as seen in Fig.2 (i.e., the SMD simulations). The accuracy of SMD can therefore largely affect the accuracy of the model's flow calculation. As seen in Fig.3 (in original manuscript), XAJ model is very good at simulating the flow variations in the Pontiac catchment with high NSE values, even during the winter season, which agrees well with the literature. This indicates the model's SMD simulations are quite realistic.

Zhou, S., Li, Y., Zhu, J., 2008. Application of Xin'anjiang model in severe cold region of Niqiu River. Water Resources & Hydropower of Northeast China, 26(9). DOI:10.3969/j.issn.1002-0624.2008.09.016

3. The authors argued that "only the surface SMD referring to the vegetation and the very thin topsoil, is utilised as a hydrological soil moisture target". Does the XAJ model allow one to simulate SMD in the "very thin topsoil"? If no, this should be clearly pointed out in the manuscript, and the simulation results' interpretation should be corrected.

Reply: XAJ model has three soil layers in its modelling structure, and yes the top layer represents the very thin topsoil. We have made this clearer in the updated manuscript.

4. I fully agree with the authors that the results are "model parameter dependent" (line 486). But I disagree that the proposed NHSMS indicators allow one to obtain independent results. I think that Figure 13 cannot be considered as an evidence of such independence because of at least two reasons: (1) only one parameter has been changed; (2) the obtained closeness of the two curves is shown for only 4 months (of 2 years) with mostly high SMD values. Thus I believe that the results presented in the manuscript are dependent on the XAJ model structure, parameters and inputs. Please give a comment.

Reply: It is true the proposed NHSMS only considers the model parameter factor, and other factors such as model structure is required in our future studies. As a result, the proposed indicator is only the very first step into creating a universal soil moisture product, and a lot of case studies adopting different hydrological models will need to be carried out. This has been pointed out in the original manuscript Lines 493-500 ("In the future it is planned to use the same process on other hydrological models to test if the normalised soil moisture indicators are not only model parameter independent but also model structure independent. Since all hydrological models are driven by the same hydrological inputs (precipitation, evapotranspiration and flow), their normalised soil moisture indicators should respond in a similar way (soil becomes wetter when it rains and drier when there is no rain). If this is true a new soil moisture product based on NHSMS could be generated as a routine product by the operational organisations such as NASA and ESA.").

The reason of showing the 4 months results is for a better visualisation. The selection of a dry period (i.e., high SMD values) is because it is the most critical period of time for the need of accurate soil moisture values for hydrological modelling. A good analogy is a hydrological model's water storage can be seen as a tea cup, during wet season with a lot of rain, it is easy to calculate how much water comes out of the cup as we know the maximum storage amount (for a hydrological model, it is via parameter calibration), so it is the total rainfall minus the maximum storage. However during dry season, it is impossible to know the outflow amount.

Therefore during the real time flood forecasting, after a long period of dryness, the accumulation of error in the hydrological models can become larger and larger with time. With accurate soil moisture information, the error could be corrected. This clarification has been added in the updated manuscript.

5. Conclusion, lines 539-541. I do not share the optimistic view of the authors on the perspective of the proposed fusion technique. Yes, the data sources contain part of useful (and probably independent) information. However, these data contain their own large measurement errors and error's synergy can result in dramatically increase of the presented results' uncertainty. I would like to read the authors' comment on this topic.

Reply: It is true that all data sources have their own errors and the errors' synergy could increase the uncertainty of the merged data. This is why a good data fusion scheme should be explored and adopted. In this study, when combining them together using the data fusion method, the total error is controlled by the desired target (the XAJ model's SMD, which has been regarded as quite realistic due to the model's good flow simulations results).

Specific Comments

1. Lines 133-136: The phrase beginning from the words "It is worth noting that. . ." looks unnecessary in scientific (non-popular) hydrological text

Reply: Agreed. It has been removed.

2. I suggest removing Fig. 3. This figure has been already demonstrated in three (at least) recently published papers of the authors (Zhuo and Han, 2016, 2017; Zhuo et al., 2015)

Reply: Agreed. It has been removed.

3. Line 165: There are 17 parameters in Table 1

Reply: Agreed. It has been corrected.

4. Eq. 1: M is not defined

Reply: The definition has been added.

5. Lines 495-496: The statement "all hydrological models are driven by the same hydrological inputs (precipitation, evapotranspiration and flow)" is misconception. Please be more precise.

[revised manuscript text omitted]

---

## Editor Decision (ED1)

I appreciate and thank the authors for their clear and concise response to feedback from all reviewers. The authors' responses were thoughtful and thorough. I find that the revised version of the manuscript is satisfactory addresses the reviewer comments and is acceptable for publication in HESS.

Alexander Gelfan
HESS Editor

| Principal Criteria | Excellent (1) | Good (2) | Fair (3) | Poor (4) |
|---|---|---|---|---|
| **Scientific Significance:** Does the manuscript represent a substantial contribution to scientific progress within the scope of *Hydrology and Earth System Sciences* (substantial new concepts, ideas, methods, or data)? | | + | | |
| **Scientific Quality:** Are the scientific approach and applied methods valid? Are the results discussed in an appropriate and balanced way (consideration of related work, including appropriate references)? | | + | | |
| **Presentation Quality:** Are the scientific results and conclusions presented in a clear, concise, and well-structured way (number and quality of figures/tables, appropriate use of English language)? | | + | | |

---

## Author Response (AR2)

**To Editor**

Dear Prof. Gelfan,

Thank you very much for the comments. The manuscript has been further amended to address the issues raised. The point-by-point replies are listed below.

If there are any further questions, please let us know.

Lu Zhuo

The Corresponding author (lu.zhuo@bristol.ac.uk )

University of Bristol

The 1st Referee expressed reasonable concern that the presented results don't allow a reader to evaluate the model's applicability because of short duration of the calibration/validation period (2 years) and suggested prolonging it. I have checked the streamflow data availability for the Vermilion River at Pontiac and found that there are at least 8-year period of continuous daily records from 2009 till 2017. Importantly, this period covers the SMOS satellite data. Thus, I strongly recommend taking into account the criticism of the referee and prolonging the study period.

Reply: We thank the editor for the careful checking. In the study catchment, there have been many data gaps from 2013-2017 (see the figures below), and the data quality in 2012 was poor. As a result, only the data in 2010-2011 are consistent and of high quality. As pointed out by Liu and Han (2010), 'Traditionally, hydrologists use rules of thumb to select a certain period of hydrological data to calibrate the models (i.e., 6 year data).' However, their study has shown 'the information content of the calibration data is more important than the data length; thus 6 month data may provide more useful information than longer data series.' Therefore, the two years of high quality data adopted in the study are better than a longer period of poor quality data. We have added this explanation to the updated manuscript to clarify the issue.

Reference:

J. Liu and D. Han, Indices for calibration data selection of the rainfall-runoff model, *Water Resources Research,* doi:10.1029/2009WR008668, 2010

[Figure]

[Figure]

[Figure]

[Figure]

[Figure]

I consider that the next comment of the referee also requires more serious attention. His/her question related to ability of the XAJ model for reproducing soil moisture dynamics under frozen conditions. This ability is important because frequent soil freezing events in winter seasons are specific for the study basin. In their response to this comment, the authors argued that "XAJ model is very good at simulating the flow even during the winter season". In my experience, this argument does not sufficient and good model's performance in reproducing streamflow does not confirm the model's performance in describing water content of frozen soil. Moreover, it is not too difficult tuning the model to short (2 years only) time-series of streamflow data without considering frozen conditions at all. Unfortunately, I could not find examples of the XAJ model's application for these conditions (the mentioned two-page paper Zhou et al, 2008 can hardly be considered as a convincing example). Thus, I recommend adding more results and discussion confirming ability of the XAJ model for reproducing soil moisture dynamics under frozen conditions

Reply:    We agree that this can be misleading. It is better to add some text based on the comments from the editor and reviewer in the updated manuscript to let the readers be aware of the issue.

--------------    to add in the manuscript ------------

[revised manuscript text omitted]